

# Explicitly modelling microtopography in permafrost landscapes in a land-surface model (JULES vn5.4_microtopography)

Noah D. Smith[1], Sarah E. Chadburn[1], Eleanor J. Burke[2], Kjetil Schanke Aas[3], Inge H.J. Althuizen[4], Julia Boike[5], Casper Tai Christiansen[6,7], Bernd Etzelmüller[3], Thomas Friborg[7], Hanna Lee[4],
Heather Rumbold[2], Rachael H. Turton[8], Sebastian Westermann[3]

[1]College of Engineering, Mathematics and Physical Sciences, University of Exeter, Exeter EX4 4QF, UK
[2]Met Office Hadley Centre, Fitzroy Road, Exeter, EX1 3PB, UK
[3]Department of Geosciences, University of Oslo, Sem Sælands vei 1, 0371 Oslo, Oslo, Norway
[4]NORCE Norwegian Research Centre, Bjerknes Centre for Climate Research, Nygårdsgaten 112. 5008 Bergen, Norway
[5]Alfred Wegener Institute Helmholtz Center for Polar and Marine Research (AWI), Telegrafenberg, 4473 Potsdam, Germany
[6]Terrestrial Ecology Section, Department of Biology, University of Copenhagen, Copenhagen, Denmark
[7]Center for Permafrost, Department of Geosciences and Natural Resource Management, University of Copenhagen, Copenhagen, Denmark
[8]UK Centre for Ecology and Hydrology, Wallingford, OX10 8BB, UK

*Correspondence to*: Noah Smith (nds211@exeter.ac.uk)

## Abstract

Microtopography can be a key driver of heterogeneity in the ground thermal and hydrological regime of permafrost landscapes. In turn, this heterogeneity can influence plant communities, methane fluxes and the initiation of abrupt thaw processes. Here we have implemented a two-tile representation of microtopography in JULES (the Joint UK Land Environment Simulator), where tiles are representative of repeating patterns of elevation difference. We evaluate the model against available spatially resolved observations at four sites, gauge the importance of explicitly representing microtopography for modelling methane emissions and quantify the relative importance of model processes and the model's sensitivity its parameters. Tiles are coupled by lateral flows of water, heat and redistribution of snow. A surface water store is added to represent ponding. The model is parametrised using characteristic dimensions of landscape features at sites. Simulations are performed of two Siberian polygon sites, Samoylov and Kytalyk, and two Scandinavian palsa sites, Stordalen and Iškoras. The model represents the observed differences between greater snow depth in hollows vs raised areas well. The model also improves soil moisture for hollows vs the non-tiled configuration ('standard JULES') though the raised tile remains drier than observed. For the two palsa sites, it is found that drainage needs to be impeded from the lower tile, representing the non-permafrost mire, to achieve the observed soil saturation. This demonstrates the need for the landscape-scale drainage to be correctly modelled. Causes of moisture heterogeneity between tiles are decreased runoff from the low tile, differences in snowmelt, and high to low-tile water flow. Unsaturated flows between tiles are negligible, suggesting the adequacy of simpler water-table based models of lateral flow in wetland environments. The modelled differences in snow depths and soil moistures between tiles result in the lower tile soil temperatures being warmer for palsa sites. When comparing the soil temperatures for July at 20 cm depth, the difference in temperature between tiles, or 'temperature splitting', is smaller than observed (3.2 vs 5.5°C). The mean temperature of the two tiles remains approximately unchanged (+0.4°C) vs standard JULES, and lower than observations. Polygons display small (0.2°C) to zero temperature splitting, in agreement with observations. Consequently, methane fluxes are near identical (+0 to 9%) to those for standard JULES for polygons, though can be greater than standard JULES for palsa sites (+10 to 49%). Through a sensitivity analysis we identify the parameters resulting in the greatest uncertainty in modelled temperature. We find that at the sites tested, varying the parameters can result in the modelled July temperature splitting being at most 0.9 or
40  3°C larger than observed for palsa or polygon sites respectively. Varying the palsa elevation between 0.5 and 3 m has little effect on modelled soil temperatures, showing that having only two tiles can still be a valid representation of sites with a large variability of palsa elevations. Lateral conductive fluxes, while small, reduce the temperature splitting by ~1°C, and correspond





to the order of observed lateral degradation rates in peat plateau regions, indicating possible application in an area-based thaw model.

## 1 Introduction

The permafrost carbon feedback is estimated to be equal to around 2 to 10% of our anthropogenic emissions budget under the Paris agreement (Burke et al. 2017; Comyn-Platt et al. 2018; Gasser et al. 2018; Koven et al. 2015; Hoegh-Guldberg et al. 2018, using the 50[th] percentile). However, the future fate of permafrost carbon depends on a complex interplay of processes, many of which are not currently included in Earth System Models. Part of the problem is a question of scale. While the extent of permafrost is vast, processes on the scale of meters compound to create a highly heterogeneous landscape, where different soil moistures and temperatures exist side by side. These processes are often driven by microtopography. Hollows trap snow leading to greater insulation in winter (Gouttevin et al., 2018). Larger snow depths are also found where snow accumulates on the upwind slope of raised areas, which is known to have a controlling effect on the growth of palsas and the presence of permafrost (Seppälä, 1994). Elevation difference also results in hollows being wetter and more conductive in the summer, in addition to having a higher heat capacity. If a pond forms, this can increase the absorption of radiation due to its lower albedo (Lin et al., 2016). Even shallow waterbodies (< 1 m) can raise the mean annual sediment temperature by more than 10°C compared to mean annual soil temperatures, and have been estimated to increase ground warming under a general climatic warming by a factor of 4 or 5 (Langer et al., 2016). The warmer, wetter areas then experience deeper thaw and increased anaerobic decomposition, producing more methane (Sachs et al., 2010). The ease with which methane can be transported out of the soil while avoiding oxidation is in turn controlled by the thickness of the soil oxic layer and the plant communities present (Lai, 2009). Again, these are things which are influenced by small scale variations, for which microtopography is a key controlling factor (Cooper et al., 2017). While methanogenesis is slower relative to aerobic respiration and the release of $CO_2$, the high Global Warming Potential (GWP) of methane means the former can be of similar importance in the short term.

The Joint UK Land Environment Simulator (JULES) (jules.jchmr.org) is a community land surface model, and is used as the land surface scheme in the UK Earth System Model (UKESM) (Sellar et al., 2019). JULES currently calculates methane fluxes using the gridcell average values of soil temperature and carbon pools for each layer (Clark et al., 2011; Gedney et al., 2004). The flux is then multiplied by a 'wetland fraction' calculated by TOPMODEL, based on the position of the water table and the gridcell topographic index (Gedney and Cox, 2003) (Figure 2). However, the 'wetland fraction' of the gridcell is likely to be physically and biogeochemically different to the rest of the gridcell. In particular, the sub-grid processes already described can result in these wetter areas also being warmer, meaning that the modelled methane flux may be too low. The resolution of these sub-grid processes and the resulting heterogeneity is similarly a key challenge for other global models used for future climate projections, and needs to be addressed in order to avoid underestimating the permafrost carbon feedback (Bridgham et al., 2013; Blyth et al., 2021; Aas et al., 2019). This study therefore aims to investigate the hypothesis that explicitly modelling microtopography is essential to accurately modelling the ground moisture, temperature and hence methane emissions of a permafrost landscape.



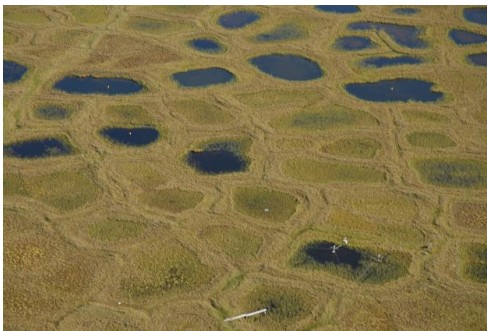 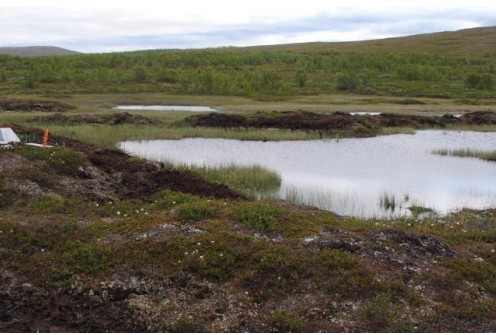

**Figure 1:** Ice-wedge polygons at Samoylov (left, J. Boike) and palsas and mire at Iškoras (right, S.E. Chadburn). Both landforms form due
to ground ice and show repeating patterns of elevation difference. Palsas have a larger elevations (0.5 - 3 m) than polygon rims (up to 1m).
Microtopography can result in wetter areas in hollows which has a warming effect in summer through the increased thermal conductivity of
saturated soil. Ponding can also have a warming effect by lowering the surface albedo. In winter wind-blown snow gathers in hollows and
provides extra insulation. The combined effect is that lower areas tend to be warmer and wetter, which in turn results in greater emissions
of methane, and permafrost thaw in those areas.


Microtopographic features are highly prevalent in the permafrost region, as they can be generated as a result of small-scale
processes through the formation or removal of ground ice. While our methods are generally applicable, this study applies our
model to two such features, ice-wedge polygons and palsas (Figure 1). Ice wedge polygons are created where wedge ice forms
in cracks created by the thermal contraction of the ground, pushing up the adjacent ground (Lachenbruch, 1962). Polygons are
widespread; wedge ice is thought to be ~20% or more of the uppermost permafrost volume (Liljedahl et al., 2016), and are
found in the continuous permafrost zone. Palsas are formed where a localised cold patch of ground causes frost heave, elevating
mounds of peat (Seppälä, 1986). These in turn experience shallower snow cover and further heaving. When these mounds are
joined to cover a larger area, they are known as peat plateaus. Palsas / peat plateaus are found in the discontinuous and sporadic
permafrost zones, and are also widespread, though not to the extent of polygons (Martin et al., 2019; Kremenetski et al., 2003;
Fewster et al., 2020). While it is hard to get an estimate for the total extent of palsas and peat plateaus, permafrost peatlands
are estimated to cover 1.7 million $km^2$ (Hugelius et al., 2020). The choice of these two landforms affords the opportunity to
test our model in climatically distinct regions and using different configurations.

Microtopography can preserve permafrost, but, conversely, can cause localised areas of raised ground temperature or 'hotspots'
where abrupt-thaw events can be initiated. These hotspots can experience slumping due to ground ice thaw, extending and
exacerbating the hotspot and possibly forming or extending a waterbody - a thermokarst lake (Langer et al., 2016). These
abrupt thaw events expose new carbon to decomposition and could have a large contribution to the permafrost carbon feedback,
potentially doubling the radiative forcing from carbon fluxes as result of gradual thaw (Walter Anthony et al., 2018). Such
abrupt thaw features are common, with thermokarst lakes covering an estimated 11% of Siberian Yedoma and much of
Northern Canada (Walter et al., 2006). However, even rarer types of abrupt thaw events are hypothesised to have a large
impact. Thaw-related hillslope erosional features are projected to affect a tiny 3% of abrupt thaw terrain, but be responsible
for a third of abrupt thaw emissions (Turetsky et al., 2020). Explicitly modelling microtopography could therefore be important
as a prerequisite to the inclusion of abrupt thaw features and their effect on the permafrost carbon feedback in land surface
models.


Modelling efforts have been hampered by a lack of long-term high-resolution observations. Nevertheless, at a handful of field
sites the local variability has been well quantified and at these sites 'tiling' approaches to modelling micro- and meso-
topography have been tested as a route to representing sub-grid processes in ESMs (Langer et al., 2016; Aas et al., 2016, 2019;
Nitzbon et al., 2019, 2020a; Cai et al., 2020; Martin et al., 2021). These authors have primarily focused on the future evolution





of permafrost landscapes, for which they have shown that including micro- and even meso-topography is essential. When applied on the global scale, a more top-down rates-based approach, similar to that taken by (Turetsky et al., 2020; Schneider von Deimling et al., 2015) may well be necessary due to the complexity of the feedbacks involved. However, this results in a loss of flexibility and predictive power, which could result in the omission of potentially important feedbacks. We therefore believe that an explicit tiling approach should be further pursued and developed. In this paper, we therefore follow the approach

of Aas et al. (2019) and Nitzbon et al. (2019) and construct a scheme within JULES where two interacting tiles, offset by an elevation difference, are taken as representative of a repeating microtopographic unit. The relative areas within this repeating unit are then representative of the entire grid-cell (Figure 2). The two tiles interact via exchange of both soil moisture and heat, by surface run-on and by redistribution of snow. A surface water store is added to the low tile to represent ponding. The two-tile approach was chosen as the simplest configuration that still includes an explicit representation of microtopography, as our

focus is testing this first step-up from the previous implicit microtopography of JULES. For the same reason, dynamically changing areas and elevations, necessary for modelling abrupt thaw processes, as well as a thorough treatment of waterbodies, though important, are not included in the scope of this investigation. Rather, this work aims to strengthen the foundations for applying tiling approaches to microtopography in permafrost landscapes in three main ways. Firstly, by evaluating the efficacy of a two-tile model against microtopographically-resolved observations of snow depth, soil temperature and moisture at four

sites. Secondly, by evaluating the importance of explicitly modelling microtopography with regards to its effect on the modelled methane emissions. Thirdly, by quantifying the effect of individual model processes, the model's sensitivity to individual model parameters and the resulting uncertainty in the modelled summer soil temperature as a result of uncertainty in those parameters. This sensitivity testing will enable future modellers to assess which model processes should be focused on, which could be simplified, and which parameters most need to be constrained.

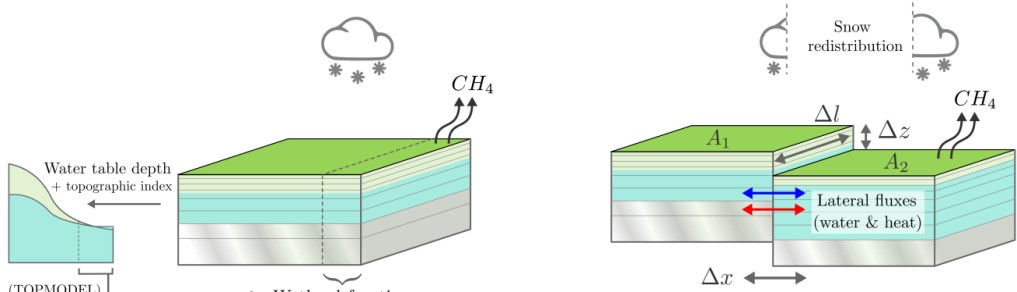


**Figure 2:** The implicit microtopography of standard JULES (left) vs our explicit approach (right). In standard JULES (left), the gridbox average values of soil temperature and carbon are used to calculate a methane flux. This is then multiplied by a wetland fraction, which is calculated by TOPMODEL using the modelled water table depth and the gridcell topographic index. Our explicit approach (right) uses two soil tiles offset by an elevation difference to represent repeating patterns of microtopography. Model spatial parameters are shown in grey

and are defined for individual sites in Table 2.

## 2 Methods

### 2.1 Sites

Four sites were used for model validation: two Siberian polygon sites, Samoylov and Kytalyk, and two Scandinavian palsa sites, Stordalen and Iškoras. This choice facilitates comparison, as sites for each type of landform have similar climate, while

polygon sites are colder and drier than palsa sites. Table 1 shows a summary of climatic variables for each site and further site information can be found in Table A 1 (Appendix A). Data used from each site can be found in Figure A 1 and Table A 2 (Appendix A). Samoylov, Kytalyk and Stordalen have already been set-up and validated in JULES (Chadburn et al., 2017) and all of these sites have microtopographically resolved data.



**Table 1:** Summary of sites. Further detail, and sources, can be found in Table A 1 (Appendix A). Acronyms - MAAT: Mean Annual Air Temperature, MAP: Mean annual precipitation.

| Site | Samoylov | Kytalyk | Iškoras | Stordalen |
|------|----------|---------|---------|-----------|
| Landform | Polygons | Polygons | Palsas | Palsas |
| Location | Lena River Delta, Siberia 72.22°N, 126.467°E | Sakha, Siberia 70.83°N, 147.485°E | Northern Norway 69.3003°N, 25.3460°E | Abisko, Northern Sweden 68.35°N, 18.817°E |
| MAAT | -12.5°C | -13.6°C | -2.1°C | -0.6°C |
| MAP | 125 mm | 202 mm | 339 mm | 305 mm |
| Thaw depth | 0.41 to 0.57 m | 0.2 to 0.3 m dry 0.4 to 0.5 m wet | 0.4 to 0.65 m for stable permafrost | 0.5 m palsas 1 to 3 m mire |
| Organic layer | 0 to 0.15 m dry < 0.2 m wet | 0.1 to 0.15m wet | 1.5m | 0.5 to > 1 m peat on Palsa |

Samoylov and Kytalyk are sites with ice-wedge polygons, where observations of soil moisture and temperature used in this paper are for low-centred polygons. Samoylov island is situated in the Lena river delta (Boike et al., 2019). Measurements are

from the Holocene terrace on the Eastern part of the island. The soil is characterized by alluvial deposits (65%), with the organic layer underlain by sand / silt with some peat layers. A shallow gradient in the terrace leads to a gradient in hydrological conditions, and consequently variation in the degradation of polygons (Nitzbon et al., 2020a). Areas of high and low-centered polygons can be seen, along with a large areal fraction of water surfaces (25%, Muster et al., 2012). Kytalyk is in a region of Yedoma, with Pleistocene ice deposits occurring nearby in 20 to 30 m tall hills (van der Molen et al., 2007). The measurement

site itself is in a depression which was originally a Holocene thermokarst lake. 2 to 3 m below this is the current river plain, where active thermokarst features are present alongside the river. In addition to ice-wedge polygons, Kytalyk has areas of small palsa-like hills (Parmentier et al., 2011). The elevation differences caused by frost-heave have led to general difference between dryer elevated areas and lower flooded areas, however ponding is only seen in small areas (up to 10%) where ice wedges are actively developing. Precipitation is smaller than that of the palsa sites, but evaporation is limited to the short

growing season, meaning the soil can remain wet.

Iškoras and Stordalen are sporadic permafrost sites with palsas surrounded by a lower-elevation, non-permafrost mire. Both have been experiencing lateral degradation of palsas. Stordalen has experienced warming of 2.5°C between 1913 and 2006 (Callaghan et al., 2010), and has lost up to 10% of the palsa between 1970 and 2000 (Malmer et al., 2005). Stordalen has areas

of palsa, ombotrophic bog which receives runoff from the surrounding palsa, and fen. Drainage out of the palsa follows the frost table via narrow 1 to 2 m channels (Olefeldt and Roulet, 2012). Unlike the palsa and bog, the fen has no permafrost. In addition to receiving drainage from the bog, the fen receives a large amount of water from a lake just outside the peatland complex, and therefore the 1.7km$^2$ catchment surrounding it. The Iškoras peat plateau is small, covering an area of ~4 ha, and is surrounded by and interspersed with fen and ponds. Similar drainage channels are observed in palsas, along with typical

signs of palsa degradation such as cracking and blocks of peat detaching from palsa edges (Martin et al., 2019).

### 2.2 JULES

JULES (jules.jchmr.org) is a community land surface model (in Best et al. (2011) and Clark et al. (2011)). It is the land surface scheme used in the UK Earth System Model (UKESM) (Sellar et al., 2019) and can also be used as an offline model, both globally and at the site level as is done here. For this study, version 5.4 (http://jules-lsm.github.io/vn5.4/, last access 2$^{nd}$ June

2020) was modified. In JULES a gridcell has a single soil column, on which surface tiles representing different plant functional





types and / or non-vegetation tiles such as bare soil, water and land ice can be used. Each surface tile models its own multi-layer snowpack and canopy water store, and fluxes of heat, water, carbon and nitrogen are aggregated before exchange with soil layers. For this study plant competition is modelled by the TRIFFID dynamic vegetation scheme. While the standard configuration of JULES usually has four soil layers with a depth of 3m, here we use a configuration with 20 soil layers with a

total soil depth of 7.8 m as suggested by Chadburn et al. (2015). JULES has been used to model permafrost at the site level, as well as its global extent and the permafrost carbon feedback, including wetland and methane emissions (Chadburn et al., 2015; Burke et al., 2017b; Comyn-Platt et al., 2018; Burke et al., 2017a; Gedney et al., 2019).

The version of JULES used here includes two tiles representing either the palsa and mire or the polygon rim and center. This
is done by repeating the same driving data twice within the model. The following developments were included within JULES to improve the representation of microtopography in the northern high latitudes. We have added snow redistribution and lateral water and heat fluxes between the tiles and a surface water store to represent ponding on the lower tile. In addition, we describe the JULES – specific modifications for this model, which are a more conceptually consistent parameterisation of the water table depth diagnostic and a new method of limiting layer numerical over- or under-saturation. Model parameters will be
defined as they are encountered but can also be found in Table 2.

### 2.3 Microtopography model additions

### 2.3.1 Snow redistribution

To model the effects of snow blown by wind on the snow depth of each tile we use the scheme of Aas et al. (2019) and Nitzbon et al. (2019) where snowfall, $S$ $(kgm^{-2}s^{-1})$ is instantaneously partitioned between the two tiles according to the current snow
depth on the tiles. We assume that while the current snow depth, $d$ $(m)$ on either tile is less than the snow catch, $s_{catch} = 0.05\,m$, the snow is 'caught' by vegetation, so cannot be redistributed. In this case, each tile experiences the same snowfall. Snow above this depth is not caught, so if the level of snow on the adjacent tile is lower, snowfall will be instantly redistributed to it. A buffer, $s_{buffer} = 0.03\,m$ is applied, such that redistribution does not happen if the elevations differ by less than this amount. This means that once the snow is approximately level, snow is once more able to fall on both tiles. This recreates the
observed behaviour of snow filling hollows first (Wainwright et al., 2017). Finally, as the areas of the tiles may be different, and snowfall in the model is per area, the redistributed snow is subject to an area scaling factor, $\frac{A_1}{A_2}$, where $A_1$ and $A_2$ $(m^2)$ are the areas of each tile. The full scheme is therefore:

*High to low tile redistribution:*

$$d_{high} > s_{catch}\,, \qquad d_{low} - d_{high} < \Delta z - s_{buffer} \qquad \rightarrow \qquad S_{high} = 0\,, \qquad S_{low} = S_{low} + S_{high}\frac{A_1}{A_2} \qquad (1)$$

*Low to high tile redistribution:*

$$d_{low} > s_{catch}\,, \qquad d_{low} - d_{high} > \Delta x + s_{buffer} \qquad \rightarrow \qquad S_{high} = S_{high} + S_{low}\frac{A_2}{A_1},\ S_{low} = 0 \qquad (2)$$

where $\Delta z$ $(m)$ is the elevation difference between tiles, and the subscripts 'high' and 'low' specify the tile with which the variable is associated.

### 2.3.2 Lateral water fluxes

Lateral flows of water were introduced into JULES using an approach mirroring the existing calculation of vertical fluxes, where fluxes are calculated based on the difference in matric potential between soil layers due to their level of saturation.
Existing functions for calculating the layer matric potentials and hydraulic conductivities are used, and the fluxes interfaced

into the existing code for the water balance for each layer. Unlike for the vertical fluxes, no implicit correction is used, as fluxes are assumed to be relatively small. Here, fluxes are calculated for horizontal connections, where connections occur between a layer and any other layers horizontally adjacent to it, taking into account the area of overlap of each connection (Figure 3). However, a 'sloped' flow scheme where layers are sequentially connected to their corresponding layer in the

neighbouring tile was also considered. Further discussion on the strengths and weaknesses of the two schemes can be found in the supplementary material. In JULES, the vertical water flux between layers per land surface area, $W_{vertical}$ ($kgm^{-2}s^{-1}$), is calculated using Darcy's law and is of the form:

$$W_{vertical} = -K_h \left( \frac{\Delta\psi_m}{\Delta z_l} + 1 \right), \tag{3}$$

where $K_h$ ($kgm^{-2}s^{-1}$) is the hydraulic conductivity, $\Delta z_l$ ($m$) is the distance between the centres of the layers, and $\Delta\psi_m$ ($m$)

the difference in matric potential between layers. When flows are horizontal, the matric gradient instead uses the horizontal distance, $\Delta x$ ($m$), and as the gravitational potential, $\psi_g$ ($m$), remains constant, $\frac{\Delta\psi_g}{\Delta x} = 0$ and the 1 can be dropped. As with the snow scheme, we must go from the intensive form used by JULES of fluxes per land surface area, to the extensive form of $kgs^{-1}$ through the interface, and back again. The calculated flux is therefore multiplied by area of the interface, $\Delta v \Delta l$, to go from $kgm^{-2}s^{-1}$ to $kgs^{-1}$, where $\Delta v$ ($m$) is the vertical overlap (orange in Figure 3), and $\Delta l$ ($m$) is

the contact length. This divided by the area of the tile that is being considered, $A_1$ or $A_2$, so that the change in water content of a layer is still expressed per unit area of that tile. The areal scaling factor for tile $n = 1$ or 2, $f_{q,n}$ is therefore:

$$f_{q,n} = \frac{\Delta l \Delta v}{A_n}, \tag{4}$$

where $A_n$ ($m^2$) denotes the area of tile $n$. The lateral flow for a connection per surface area of tile $n$ $W_{horizontal,n}$ ($kgm^{-2}s^{-1}$) is then calculated as:

$$W_{horizontal,n} = -K_h f_q \frac{\Delta\psi_m}{\Delta x}, \tag{5}$$

In JULES, $\psi_m$ and $K_h$ can be calculated using either the Brooks and Corey (BC) or the Van Genuchten (VG) relations (see Best et al., 2011). Here we use the BC relation for $K_h$. The VG relation is used for $\psi_m$, as it avoids a mismatched $\psi_m$ when both layers are saturated but is otherwise almost identical to the BC relation for the soil properties of the modelled sites. As in the case for the vertical fluxes, the conductivity is determined using the average liquid water content of the connected

layers. This calls into question the use of only two boxes, as the sloped area of polygonal tundra has been observed to have a lower saturation than either the polygon rim or centre (Boike et al., 2019) which could lead to the flow being restricted. Not modelling the slope is however similar to previous approaches (Aas et al., 2019; Nitzbon et al., 2019) and is the simplest configuration that still models microtopography.

The horizontal flow scheme raises the question of what to do with the upper layers of the raised tile, which may have no horizontally adjacent layer. If the water table in the elevated tile is above the surface of the lower tile, and above the level of the surface of any ponded water (if present), water will be able to laterally egress the soil (Figure 3 C). Similarly, if a pond is present on the low tile and the surface of the pond is above the level of the water table in the high tile, water can flow from the pond laterally into the soil (Figure 3 D). For these flows it is therefore necessary to determine the level of the local water table.

For the high tile, for a particular connection, the local pressure head in the high tile $\psi_{ph}$ ($m$) is the height of the water table above the midpoint of the vertical overlap (orange in Figure 3) of the connection being considered. The overlap and midpoint of the connection may be different to the *layer* thickness and midpoint. This could be due to the layer being partially saturated (the highest overlap in Figure 3 C, and the lowest overlap in Figure 3 D), if only part of the layer is above the pond height (the lowest overlap in Figure 3 C) or if the pond height is between the lower and upper boundary of the unsaturated layer (the

highest overlap in Figure 3 D). To calculate the water fluxes for water egress, the difference in potential $\Delta\psi = \psi_{ph}$, as there



is no matric suction from air. For water ingress, $\Delta\psi$ is the height of the pond above the midpoint of the connection, plus the matric suction of the layer the water is entering.

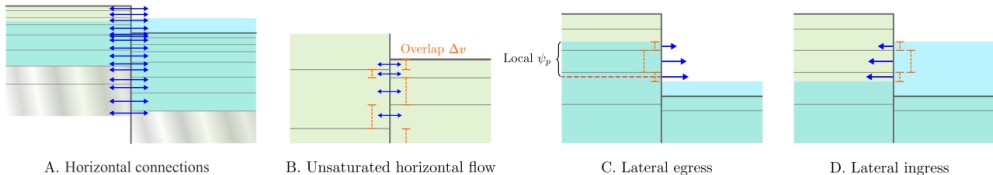

A. Horizontal connections    B. Unsaturated horizontal flow    C. Lateral egress    D. Lateral ingress

**Figure 3:** The lateral flow scheme. Fluxes are calculated horizontally for connections where layers overlap, and in the same manner as for the vertical fluxes in JULES. Fluxes are only permitted when they are into an unsaturated layer. Where high tile layers are above the low tile soil surface, flow can occur horizontally out of the soil, or from the pond to the soil.

While the local pressure head must be calculated in order for horizontal flows between the soil and the air or the pond and the soil to be modelled, flows are always from a saturated or partially saturated region to an unsaturated one. The model

therefore does not implement saturated lateral flows, which are conceptually important for flows between polygons (Wales et al., 2020). This means that advective flows of heat may not be properly represented. However, the model will still be able to act to balance the water table and moisture potentials between tiles, though the rate with which equilibrium is reached may be different. Any discrepancy in rate is however expected to be less than the usual timescales over which the water table changes and will therefore not be a problem. The addition of saturated-saturated flows would require solving for the pressure

potentials of multiple saturated layers of varying soil properties, which may or may not contain frozen water. This is non-trivial, as the pressure potentials themselves depend on the (now 2D) water fluxes, leading to the requirement for a more complex iterative scheme. On balance, we consider this scheme to be a reasonable simplification for our purposes.

### 2.3.3 Lateral heat fluxes

Lateral fluxes of heat by conduction and advection use the same connections as the water fluxes. However, of these, only the

soil to soil connections are used, as the pond thermodynamics are not explicitly modelled (it is assumed to be at the surface temperature). The equations for lateral heat flows have the same form as those for the vertical fluxes between layers, as before with the addition of the appropriate areal scaling factors, namely:

$$G_n = -\lambda_{soil} f_{n\,q} \frac{\Delta T}{\Delta x} , \tag{6}$$

$$J_n = -c_w W_{h,n} \Delta\text{T} , \tag{7}$$

where $G_n$ $(Wm^{-2})$ and $J_n$ $(Wm^{-2})$ are the conductive and advective heat flows for a lateral connection per surface area of the tile respectively. As before, $n$ refers to the tile that is being considered. $W_n$ is the lateral water flow from the previous section, $\Delta T$ $(K)$ is the temperature difference between laterally adjacent layers and $c_w$ $(Jkg^{-1}K^{-1})$ is the specific heat capacity of water. The average thermal conductivity of the two soil layers $\lambda_{soil}$ $(Wm^{-2}K^{-1})$, is calculated using the usual JULES function (Best et al., 2011), using the mean of the frozen and unfrozen water contents and of the dry soil thermal

conductivity for the two layers being considered.

In reality, the heat flux is determined by the continuous derivative $\partial T/\partial x$. The temperature profile could vary horizontally in a different way to the moisture profile, and indeed would be expected to be asymmetrical across a frozen-unfrozen interface. In the future, the need for $\Delta x$ could be eliminated by dynamically determining this profile and including lateral

freezing/thawing at the interface. For now, in the model freezing and thawing are effectively distributed evenly throughout the layer undergoing phase change and we use the same $\Delta x$ as in the water flux calculation.





### 2.3.4 Ponding and run-on

In the studied sites, the lower area tends to remain almost saturated year-round (Figure 6 & Figure 8). However, in standard JULES the surface layers rarely become saturated, as after rainfall events or snowmelt water quickly infiltrates into a space

provided by evaporation, or, as happens after snowmelt, water is in excess of the maximum infiltration rate and runs off. At both types of site, ponding is common (Boike et al., 2013; Klaminder et al., 2008), both in the centres of polygons and around the edges of palsas. This enables water to gradually infiltrate rather than running off and provides a buffer that enables the soil surface to remain saturated. To model this, a surface water store was added to the low tile. At each timestep, a fraction, $f_{ro}$, of runoff from the high tile joins the throughfall of the low tile as run-on, along with the amount of water in the surface water

store. Water in excess of the maximum infiltration is then stored in the surface water store. When a pond is present, bare soil evaporation is switched off, and evaporation from the surface water store is equal to the potential evaporation rate calculated by JULES (Best et al., 2011). Currently, no thermal effects of the pond are included, and the pond cannot freeze, though in future aspects of FLake (Rooney and Jones, 2010) could be used to introduce these processes, in a similar manner to (Langer et al., 2016). The ability to pond improves the surface wetness, and if the surface is able to become saturated year-round, the

pond can build up to become a persistent waterbody (Figure 6). When not limited, even in the wettest configuration, pond depths did not rise to more than a couple of metres. However, as this may not physically possible due to the size of the hollow, ponded water in excess of a fraction $f_{pd}$ of the elevation of the high tile is removed as runoff.

### 2.4 JULES–specific modifications

#### 2.4.1 Diagnosing the water table position in a partially saturated layer

The implemented lateral flow scheme requires the position of the local water table for a layer to be determined, here referred to as the 'layer head', or $\psi_p$. Here, we present a new method of determining the water table within a partially saturated layer in a way that is consistent with the assumptions of the cell-centred numerical method for solving the Richards equation used by JULES. The model procedure for finding the layer head is then described. As mentioned in the previous section, JULES calculates the matric potentials of layers using the fractional saturation of the layer, $\theta$. Previously, the position of the water

table in JULES has been calculated by assuming that $\theta$ refers to the saturation at the midpoint of the layer, $\theta_{mid}$. This aligns with the usual calculation of vertical Darcy fluxes using the potential difference between the midpoint of each layer. Setting the potential of the water table to be 0, the soil moisture above the water table is assumed to have the equilibrium profile such that the increase in gravitational potential is balanced by a decreasing matric potential such that the total potential is constant. Using the Brookes-Corey (BC) relation at the layer midpoint,

$$\psi = \psi_s \theta_{mid}^{-b} = z_w \, , \tag{8}$$

where $z_w$ $(m)$ is the height above the water table, $\psi_s$ $(m)$ is the BC matric potential at saturation and $b$ is the Clapp and Hornberger (1978) soil exponent. The result of this calculation is shown by the blue curve in Figure 4. A similar calculation can be done using the Van Genuchten (VG) relation, for which we use an alternative formulation:

$$\psi = \psi_s \theta_{mid}(\theta_{mid}^{-b-1} - 1) = z_w \, , \tag{9}$$

with similar results to BC, giving the orange curve. This correctly places the water table at the midpoint when saturated, rather than being offset by $\psi_s$.





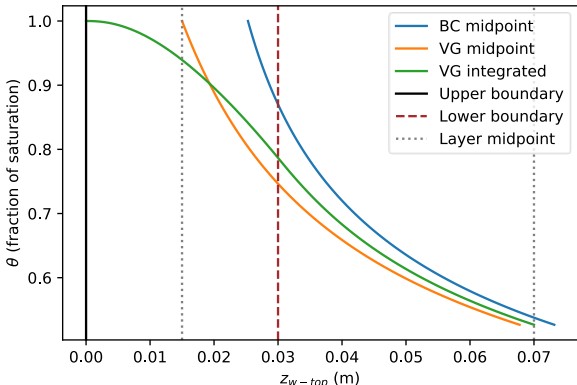

**Figure 4:** The fractional saturation of the layer, $\theta$, vs the distance of the water table below the top of the layer $z_{w-top}$. Previously, the position of the water table has been calculated by assuming that $\theta$ refers to the fractional saturation at the midpoint of the layer, resulting in the blue and orange curves, and the water table not exceeding the midpoint of the layer when the layer is saturated. However, in JULES, when considering how much water fits in a layer, $\theta$ refers to the average saturation of the layer. Using this interpretation results in the green curve, where the water table is at the top of the layer when the layer is saturated. Making the simplification that the green curve is approximately linear while the water table is within the layer, the position of the water table within a partially saturated layer can easily be found from the average saturation of the layer.

However, in JULES, $\theta$ is also a prognostic variable tracking the average saturation of a layer and is updated according to how much water flows into or out of a layer. This contradicts the calculation of the water table depth and of the direct calculation of the matric potential at the midpoint from $\theta$, as a saturated layer ($\theta = 1$) would have its water table at the top, and conversely the saturation at the midpoint could be 1 even when $\theta$ is not. We therefore need a way of calculating the water table from the value of the average saturation of the layer. We start by doing the reverse. If the water table distance below the top of the layer, $z_{w-top}$ (m), is known, the average fractional saturation of the layer, $\theta_{avg}$, can be found by dividing the integration of the equilibrium VG profile for $\theta$ over the layer by the layer thickness, $z_T$ $(m)$, noting that below the water table is saturated.

$z_{w-top} <= z_T$:

$$\theta_{avg} = 1 - \frac{z_{w-top}}{z_T} + \frac{1}{z_T} \int_{z=0}^{z_{w-top}} \left( \left( \frac{z}{\psi_s} \right)^{\frac{b+1}{b}} + 1 \right)^{\frac{1}{-b-1}} dz , \qquad (10)$$

$z_{w-top} > dz$:

$$\theta_{avg} = \frac{1}{z_T} \int_{z=0}^{z_{w-top}} \left( \left( \frac{z}{\psi_s} \right)^{\frac{b+1}{b}} + 1 \right)^{\frac{1}{-b-1}} dz , \qquad (11)$$

This gives the green curve in Figure 4. The water table is now at the top of the layer when the layer is saturated. The deviation from either the standard VG or BC calculation is small when the water table is below the bottom boundary of the layer, indicating that $\theta_{avg} \approx \theta_{mid}$ is a good approximation for most cases away from the saturated/unsaturated boundary. Accounting for the difference is therefore only of any importance in wetland environments. To invert the function to calculate the water table depth from $\theta_{avg}$ is non-trivial. However, within the ranges of $b$ and $\psi_s$ used at the studied sites, $z_{w-top}(\theta_{avg})$ is approximately linear while the water table is within the layer. We therefore calculate $\theta_{min}$, the value of $\theta_{avg}$ when the water table is at the bottom of the layer ($zw_{top} = z_T$) and interpolate between ( $\theta_{min}, z_T$) and (1,0) to finally find $z_{w-top}$ as a function of $\theta_{avg}$:

$$\theta_{z_T} = \left( \left( \frac{z_T}{\psi_s} \right)^{\frac{b+1}{b}} + 1 \right)^{\frac{1}{-b-1}} , \qquad (12)$$



$$VG_{int}(\theta) = \int \psi_s \theta (\theta^{-b-1} - 1)\, dz = \psi_s \left( \frac{1}{-b+1} \theta^{-b+1} - \frac{\theta^2}{2} \right), \tag{13}$$

$$\theta_{min} = \frac{1}{z_T} \left( VG_{int}(1) - VG_{int}(\theta_{z_T}) \right) + \theta_{z_T}, \tag{14}$$

when $\theta_{avg} < \theta_{min}$:

$$z_{w-top} \approx \frac{z_T(\theta_{avg} - \theta_{min})}{1 - \theta_{min}} \tag{15}$$

The final method is simple to implement and compares well to the full numerical integration ('VG integrated' in Figure 4), with fully accurate end-point values.

Now that the position of the water table within a partially saturated layer can be found, the model procedure for finding the local water table for a layer is as follows: if a layer is saturated, then each layer above the original layer is checked sequentially

until a layer that is partially saturated is found. The position of the water table within this partially saturated layer is then determined, and the total height from the original layer to this is the layer head. If the original layer is partially saturated, the position of the water table within the original layer is determined only if the layer beneath is saturated. If a layer has frozen water present, then the layer head is zero for that layer. If, while sequentially checking the layers above the original layer, a layer which contains frozen water is found, the layer head is the height from the original layer to the bottom of that layer. If

the saturation continues to the surface, then the layer head is the height from the original layer to the pond surface (if present).

### 2.4.2 Limiting over- or under-saturation

The standard methods JULES uses to avoid supersaturation or undersaturation as a result of the water flux calculation numerics can result in the unintended consequence of water being passed out of the soil column. This is particularly a problem for freezing saturated soils. Here, we present a method which avoids this problem, and which also integrates with the scheme for

simulating lateral fluxes of water.

In some cases, the explicitly and implicitly calculated Darcy fluxes could lead to layers which have a fractional saturation outside the range of 0 to 1. To avoid this, JULES can be set to either limit the water flux into the top of the layer (l_soil_sat_down = false) or to limit the flux out of the bottom of the layer (l_soil_sat_down = true) so that the saturation does

not go outside this range (Best et al., 2011). The soil water extraction and sub-surface runoff are not limited, as they cannot cause oversaturation, and should be absent in the case of under-saturation. However, both approaches can cause problems, as in both cases the choice of which flux to limit does not account for the direction of water flow. This can cause water to either be able to pass downwards through permafrost, or be ejected upwards out of the soil when water is drawn upwards towards a frozen surface layer when refreezing (Figure S 1 and Figure S 3, Supplementary). In addition, an approach is needed that can

also limit the lateral fluxes when necessary.

Here, we solve this problem by only the limiting the *incoming* fluxes when a layer will potentially be oversaturated (soilsat-updown). This is done by multiplying them by an appropriate factor such that the layer becomes exactly saturated (Figure S 2, Supplementary). A multiplicative factor was used so that all fluxes scale equally. For a potentially under-saturated layer, any

outgoing fluxes (discounting qbase and extraction) are limited in a similar manner. This is necessarily an iterative process as for any layer either flux could be limited, potentially creating the need to adjust a flux in an adjacent layer. The iteration was found to be most efficient sweeping upwards from the lowest layer, and repeats until either there is no appreciable change, or the iteration limit (set at 1 greater than the number of layers) is reached. Generally, only a few of sweeps are required, and the iteration limit is never reached. As lateral flows can be limited, which changes the flows of the paired tile, the limiting code is

run a second time for each box after it has been run for all boxes once.



For the most part, soilsat-updown results in a soil moisture profile very similar to that of l_soil_sat_down = false, and there are only a very few cases where flux out of the top of the soil is avoided (Figure S 1, Supplementary). This does however suggest that l_soil_sat_down = false is in general the more physically realistic scheme. This being said, the few instances

where the top of the soil is saturated, and layers of soil exhibit an upward suction into saturated layers are of importance to freezing wetland environments and become more frequent with the inclusion of ponding. Best et al. (2011) point out that, considered globally, l_soil_sat_down = false generates too much runoff in permafrost environments due to correctly allowing a perched water table above a frozen layer. However, the perching of the water table above saturated permafrost is a key behaviour responsible for permafrost wetlands (Avis et al., 2011), so setting l_soil_sat_down =  true is undesirable. Our

addition of ponding will help reduce runoff, while the effect of macropores (Bechtold et al., 2019) should also be considered, so as to get the right infiltration in the right places for the right reasons. As a side note, an alternative, more physically realistic solution to water being drawn into already saturated frozen layers would be the inclusion of frost heave.

### 2.5 Setup of simulations

#### 2.5.1 Site parameters

The model is parameterised using observed characteristic dimensions of landscape features present at sites. These parameters are: the elevation difference between tiles $\Delta z\ (m)$, the areas of each tile $A_1$ and $A_2\ (m^2)$, the contact length between tiles $\Delta l\ (m)$ and the horizontal distance between tiles $\Delta x\ (m)$ (Figure 5). For the polygon sites, the raised and lower tiles represent the polygon rim and centre. For the palsa sites, the tiles represent the palsa and mire. The ratios of these parameters determine the model's geometry, so the model could be readily applied to other forms of microtopography also.  We have approximated

ice-wedge polygons to have a circular centre surrounded by a ring, so $A_1 = \frac{1}{\pi}(\frac{\Delta l}{2})^2$, while palsa and mire are represented by squares of equal area with one side in contact, so $A_1 = A_2 = \Delta l^2$. $\Delta x$ is chosen to be representative of the length over which the transition between the thermal and hydrological regimes of the two tiles takes place, as it is this that affects the magnitude of the fluxes.  $\Delta x$ is therefore less than the distance between the centres of each tile.

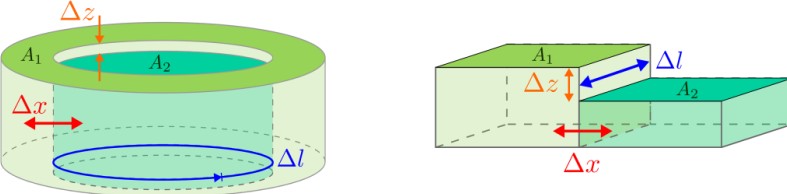

**Figure 5:** Geometries indicated by relationships between spatial model parameters for polygons (left) and palsa mires (right).

Polygon spatial parameters for Samoylov were found by measuring polygons in the Digital Elevation Map by Julia Boike et al. (2019). Averages were taken over the major and minor axes of each polygon, and these used to calculate an average area for the rim and centre. For Kytalyk, the polygon Lhc11 in (Teltewskoi et al., 2019) was used. In both cases, these are low

centred polygons. For the palsa-mire configuration, the geometry of two bordering squares was chosen as a simple configuration representing the edge of a peat plateau, this being an area of interest due to being the site of palsa degradation. In reality, the geometries of peat plateaus are complex, leading to extended perimeters. Indeed, both the perimeter to area ratio and the rate of areal degradation have been observed to increase as palsas become smaller (Borge et al., 2016). The outcome of varying $\Delta l$ will therefore be of interest for these sites. The chosen geometry for peat plateaus can however be a valid

representation if a number of assumptions are true. That the border is straight is a good approximation if the border length is small. That the tiles only exchange fluxes on one side relies on gradients of moisture and temperature being small on the other





three sides. This ought to be true of the two sides perpendicular to the border due to symmetry, and the border being small. Gradients of soil moisture and temperature will also be small at the side furthest from the border if the distance between this side and the border is great enough. Squares of side length 5 m were chosen for both sites as a compromise between these

assumptions. The model's sensitivity to this choice was tested later. The elevation used for Iškoras was taken to be the same as the elevation of the palsa where the snow depths were measured (0.68 m) so that results could be validated. This falls towards the lower end of the range for the site observed by Martin et al. (2019), namely 0.1 to 3 m, but is nonetheless an established palsa. Stordalen is a site experiencing degradation, for which Karlgård (2008) gives the characteristic height of 0.5 m, Olefeldt and Roulet (2012) give a range of 0.5 to 2 m, Klaminder et al. (2008) a range of 1 to 3 m. Due to the lack of

specific microtopographic observations for this site, an elevation difference of 2 m was chosen to provide a contrast to the model setup for Iškoras. The resulting estimates of site-specific model parameters are given in Table 2.

**Table 2:** Estimates of site-specific model parameters. S.S. min and max define the range over which the parameter was varied for the Latin Hypercube Sampling in the sensitivity study (section 2.5.2 **Configurations of JULES simulations**).

| Parameter | Description | Samoylov | Kytalyk | Iškoras | Stordalen | S.S. min | S.S. max |
|---|---|---|---|---|---|---|---|
| $\Delta z$ | Elevation (m) | 0.38 | 0.35 | 0.68 | 2.0 | 0.0 | 3.0 |
| $A_1$ | High box area (m$^2$) | 70.0 | 23.4 | 25.0 | 25.0 | 10.0 | 800.0 |
| $A_2$ | Low box area (m$^2$) | 58.0 | 19.6 | 25.0 | 25.0 | 10.0 | 800.0 |
| $\Delta l$ | Contact length (m) | 26.7 | 15.7 | 5.0 | 5.0 | 10.0 | 120.0 |
| $\Delta x$ | Horizontal distance (m) | 2.1 | 1.2 | 2.0 | 2.0 | 0.5 | 50.0 |
| | | | | | | | |
| $s_{catch}$ | Snow catch (m) | 0.05 | 0.05 | 0.05 | 0.05 | 0.0 | 0.1 |
| | | | | | | | |
| $f_{pd}$ | Max pond depth as a fraction of elevation | 0.9 | 0.9 | 1.0 | 1.0 | 0.0 | 1.0 |
| $f_{d1}$ | Drainage from high box as a fraction of that calculated by TOPMODEL | 1.0 | 1.0 | 0.0 | 0.0 | 0.0 | 1.0 |
| $f_{d2}$ | Drainage from low box as a fraction of that calculated by TOPMODEL | 0.0 | 0.0 | 1.0 | 1.0 | 0.0 | 1.0 |
| $f_{ro}$ | Fraction of runoff from tile 1 that runs-on to tile 2 | 0.9 | 0.9 | 0.8 | 0.8 | 0.0 | 1.0 |


### 2.5.2 Configurations of JULES simulations

For model code and configuration files, see the code and data availability statement. Each site was spun up in standard JULES (without any microtopographic additions or tiling) for 10,000 years using repeating driving data for the years 1901-1910. Admittedly, this length of spinup turned out to be unnecessary as methane fluxes were ultimately calculated offline using

observed carbon profiles. This spun up state with the additional modification of a saturated soil column was used to initialise the main run (1901 - 2016) for both standard and tiled JULES. As the same spun up state was used to investigate multiple configurations which may have different hydrological end states, the soil column was set to be initially saturated for both the spinup and the transient run. This was to avoid the possible development of dry layers during spinup persisting due to being frozen, which could then limit the possible end states (Figure S 5 and Figure S 6, Supplementary). The thermal and hydrological

states of the tiles of the transient runs were stable by 1911, and our analysis is limited to the 21$^{st}$ century. Forcing data were prepared as described in (Chadburn et al., 2015), where reanalysis data for the grid cell containing each site were adjusted using site observations where available. Water and global change Forcing Data (WFD) were used for the period 1901-1979 (Weedon et al., 2010, 2011) and WATCH Forcing Data Era-Interim (WFDEI) used for the period 1979-2017 (Weedon et al., 2014, 2018). Parameters for the main tiled runs are given in Table 2 & Table 3.





For the sensitivity study, three different approaches were used to test the size of the effect of different model processes and the model's sensitivity to its parameters:

- *Additive/subtractive process switching*: a series of runs where aspects of the model were switched on individually, using standard JULES as the base configuration, or switched off individually using the full tiled runs as the base configuration.
- *Latin hypercube sampling*: a series of 100 runs where model parameters were varied independently using a Latin hypercube sampling, each parameter having an even distribution between the maximum and minimum values given in the columns S.S. min and S.S. max in Table 2. These runs enable the possible range for an output variable to be gauged, with the caveat that the low ratio of samples to parameters mean that configurations where a number of parameters are at their extreme are unlikely to be sampled.
- *Individually varying parameters*: A series of runs where we varied each parameter individually using same maximum and minimum values as for the hypercube sampling, while holding the other parameters at their standard tiled configuration values (as in Table 2). These runs enabled the effects of individual parameters to be disentangled. Using these outputs and estimates of site-specific parameter uncertainties enabled the uncertainty in an output variable due to the uncertainty in a particular parameter to be found.

### 2.5.3 'Offline' methane analysis

The methane fluxes for each site were estimated 'offline' in R using the equations from the methane model in JULES applied to soil carbon and temperature data, similar to part of the analysis in S. E. Chadburn et al. (2020). The difference here was that we used a hybrid approach where observed soil carbon profiles were used in combination with modelled soil temperatures (as opposed to using only observations). We took this approach as it removes error introduced via poor simulation of soil carbon, since the soil carbon was not developed or evaluated in this study. This approach enables the impact of substrate vs temperature dynamics to be disentangled. For this work we look at methane emissions as calculated by the previous non-microbial version of JULES, as well as by the recently added microbial scheme. The microbial scheme adds a dissolved organic carbon pool, a methanogen pool, and the associated dynamics of methanogenic growth and dormancy. This has been shown to better represent the observed seasonal dynamics (S. E. Chadburn et al. 2020). Work to improve the soil carbon model in JULES is ongoing.

### 2.5.4 qbase configuration (modelling baseflow)

JULES uses TOPMODEL to calculate the baseflow (qbase) or 'subsurface runoff' for each soil layer, based on a topographic index and the position of the water table. This acts as the gridcell-scale lateral drainage. Here, we also qbase set to zero for layers which are unsaturated and/or frozen. However, in this explicit representation of microtopography, the lateral flows between tiles directly model at least part of this flow. For this reason, we include the parameters $f_{d1}$ and $f_{d2}$, which scale the drainage calculated by TOPMODEL applied to the high tile and low tile respectively. We first considered the configuration where qbase is set to zero for the simulated palsa and for the polygon centre and is on for the other tiles. This configuration is referred to as 'part qbase', or 'qbase on' as this is the main configuration where we allow qbase in the model (Table 3). The motivation for this is that there cannot be any drainage from the centres of polygons without going through the rim, and similarly drainage from palsas would usually pass through the surrounding mire. There are two main problems with this approach. Firstly, we are applying a model based on gridcell-scale assumptions to a subgrid model. This may result in qbase being too great or too small. For instance, rather than being calculated using the gridbox average saturation, it is now calculated using the saturation of either the wetter or the drier part of the landscape. This approach could also result in a reduced qbase as it is only being applied to part of the area. Secondly, the lateral drainage calculated by TOPMODEL may not be suitable for some instances of wetland, or where a large proportion of the gridcell is wetland. This is partly due to TOPMODEL's





limitations in flat landscapes (Koster et al., 2000), but is also due to wetlands being formed in areas where lateral drainage is impeded, or even which are fed by groundwater. In JULES, qbase only ever removes water, so the latter is not possible. The repeating nature of landscape units provides motivation for some degree of impedance, as two symmetrical bordering units

can have no flow between them. As we expect that qbase may be further impeded, we also consider the configuration, referred to as 'no qbase', where qbase is switched off for both tiles. This is not a general solution and is a departure from standard configurations of JULES. In reality, the lateral landscape-scale drainage will depend on the connectivity of the polygon troughs or mire network (Liljedahl et al., 2016; Connon et al., 2014), in addition to the presence of external reservoirs and inter-gridcell flows. These things are beyond the scope of this study and will require further consideration. In addition to these two main

configurations for qbase, the response of the model to varying fd1 and fd2 between 0 and 1 was tested in the sensitivity study (as described in section 2.5.2 Configurations of JULES simulations), and a configuration with qbase on for both tiles ('twin qbase') was included in the model process switching tests.

**Table 3:** Configurations of the parameters $f_{d1}$ and $f_{d2}$, which determine the drainage as a fraction of that calculated by TOPMODEL for
each tile.

| Drainage as a fraction of that calculated by TOPMODEL | Polygons | | Palsas | |
|---|---|---|---|---|
| | $f_{d1}$ (high tile) | $f_{d2}$ (low tile) | $f_{d1}$ (high tile) | $f_{d2}$ (low tile) |
| Full tiling (qbase on) / 'part qbase' | 1 | 0 | 0 | 1 |
| Full tiling (no qbase) | 0 | 0 | 0 | 0 |
| Twin qbase | 1 | 1 | 1 | 1 |

**3 Results and discussion**

An overview of the model output can be seen in Figure 6, which displays climatologies for Samoylov and Iškoras, showing
soil moisture and temperature with depth, along with snow depth and modelled pond height, comparing observations with standard and tiled JULES. These results will be expanded in the following sections, which examine the effect of the tiled model on snow, before moving on to soil moisture and temperature, and considering which model processes have most effect on these variables. The resulting effect on methane is considered, in addition to possible applications of the model. Finally, the sensitivity of the model to its parameters is investigated, and sources of uncertainty identified. Where climatologies are shown,
the years shown are for those the model is averaged over, while the years averaged over for observations will be for all years available (see Figure A 1, Table A 1).





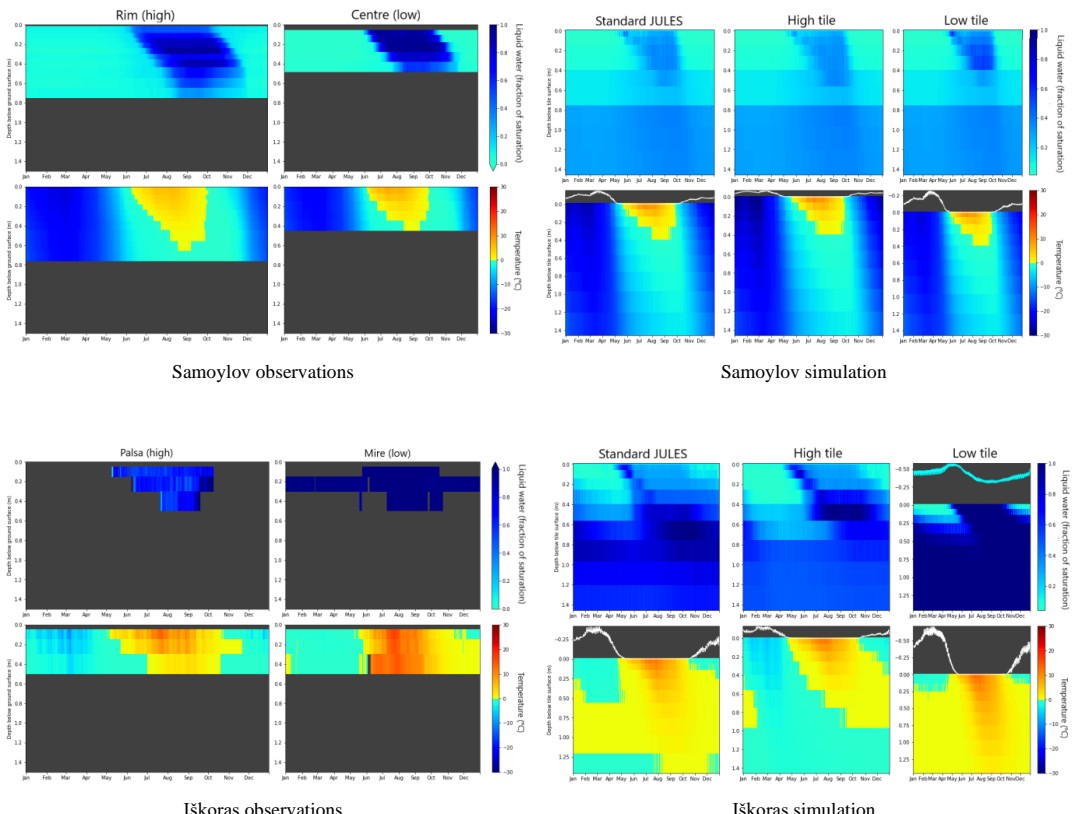

**Figure 6:** Climatologies (2002-2016) of soil moisture and temperature for the polygon site Samoylov and the palsa site Iškoras, comparing observations with standard and tiled (no qbase) JULES. The white line above the plots of soil temperature shows simulated snow depth, the cyan line above the plots of simulated soil moisture shows simulated pond depth. For Samoylov, the tiled model has little effect on thaw depths. For Iskoras, thaw depths are much greater, and while the low tile does not get as warm as observed, the tiled simulation clearly differentiates between the permafrost palsa and non-permafrost mire. Iskoras is a wetter site, and the tiled simulation is able to develop a persistent pond. The greater elevation difference at Iskoras means that the saturated low tile has less effect on the upper layers of the high tile. The tiled simulation does not develop a pond for Samoylov, however, while not shown on this figure, a seasonal pond does develop for the other polygon site, Kytalyk. Soil moisture for the low tile is improved for Iskoras vs standard JULES, however the simulation for Samoylov remains too dry.

### 3.1 Snow depths

The splitting in snow depths between the tiles as a result of the snow redistribution scheme is shown in Figure 7. When comparing the average climatologies for Iškoras, where microtopographically resolved timeseries of snow depth are available,
both the snow depths for the top and the base of the palsa are over 20 cm closer to observations for March than the standard JULES simulation. Snow depths for the polygon centre for Samoylov are similarly closer. The goodness of the model's operation mostly depends on the forcing applied, how well the typical maximum snow height on the elevated areas matches the snow catch, and the tile areas. This is because most of the time the modelled low tile snow depth for these sites does not reach the elevation difference, the point at which snow in the model would build up on both tiles at the same rate. Consequently,
the snow depth on the high tile is usually limited by the snow catch parameter ($s_{catch}$), while the low tile snowfall receives a fixed adjustment of snow for the time that the snow depth is above the snow catch. An alternate 'sloped' snow scheme was trialled whereby the fraction of snowfall redistributed varied linearly between all the snowfall being redistributed if the





difference in snow elevations equalled the elevation difference between tiles, to no snow redistributed if the snow elevations
were equal. The motivation for this scheme was that it might have been more suitable for polygon sites where the slope of the
rim gradually becomes more covered in snow as the depth of snow in the centre increases. However, there was little difference
between average snow depths for each scheme, so the 'sloped' scheme was found to be unnecessary.

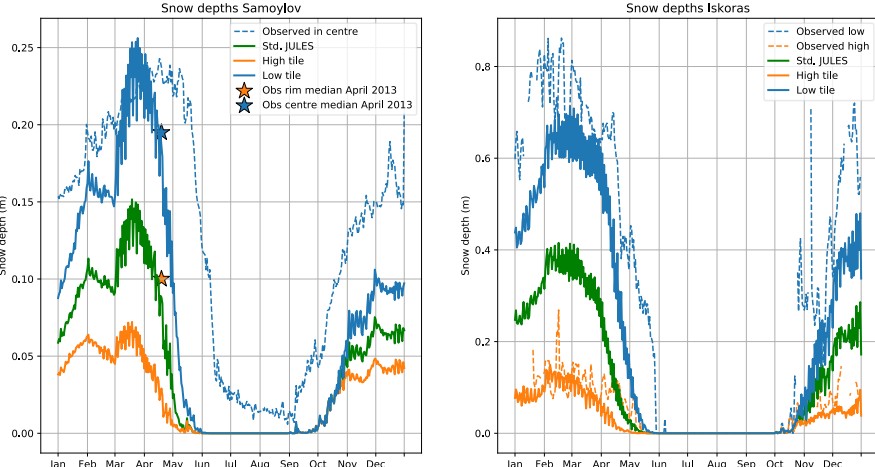

**Figure 7:** Climatology (2002-2016) of simulated vs observed snow depth for Samoylov (left) and Iskoras (right). The tiled model
generally performs well at differentiating the snow depths between the deeper snow depths in hollows vs the raised areas. The
tiling model does not correct the snow melt being too early.

At Iškoras, while the average climatologies are in relatively good agreement, it is hard to fully gauge the simulation's accuracy
as the observed snow depth time series for Iškoras do not overlap the model run period. For Samoylov, the timeseries do
overlap, but only a timeseries for the centre is available. When comparing modelled snow depths for Samoylov to spatially
resolved measurements by Gouttevin et al. (2018) on their campaign in April 2013, the simulation has the correct depth of
snow on the rim, but around 20 to 30 cm too much snow on the centre, taking into account the observed spatial variability.
Some of this discrepancy may be due to the forcing data used. For instance, in the forcing data for Samoylov there is almost
no snowfall prior to the start of October, whereas observations indicate snowfall before this, resulting in around 5cm too little
depth of snow for October to January. The scheme also has some obvious omissions. While snow is assumed to be redistributed
by wind, wind speed and direction is not input into the snow scheme. Spatial variability due to wind direction and drifting
against slopes is therefore not considered. Much of the snow can also be blown a greater distance and accumulates in larger
depressions such as lake basins and river gullies, therefore in reality the total balance of snow needs to be considered on a
much larger scale. However, a more complete accounting for the spatial variability due to effect of wind speed and direction,
topography and vegetation, as in SnowModel (Liston and Elder, 2006), moves towards needing a detailed gridded
representation of topography to run, and goes against our aim for a simple representation of subgrid processes. Currently, the
snow catch is unaffected by the type and height of vegetation present in the model. Linking the snow catch and vegetation
height would be a key next step and would enable snow-vegetation feedbacks to be investigated. Lastly, the properties of the
falling snow have no effect on the modelled redistribution, and no consideration is given to wind erosion of the snowpack at a
later time after the snow has settled. Zweigel et al., (2021) included the properties of falling snow and the effects of internal
snow processes and snow microstructure on snow mobility for a high-Arctic site on Svalbard. They found that a thinner layer
of lower density snow built up on their high 'ridge' tile than their lower 'snowbed' tile, which was then able to be subject to
wind erosion at a later time. However, while including these omissions would affect the properties, timing and spatial
variability of the modelled snow redistribution, the snow depths on each tile have still been successfully differentiated with



the current model. Collection of additional spatially resolved time series data would enable further validation, investigation of
the effect of missing processes and for model parameters to be better constrained.

### 3.2 Water

Figure 8 shows in black the observed average saturation profiles for July, where sites are grouped according to landscape type
due to a lack of detailed observations of hydrology at Kytalyk and Stordalen. For all studied sites, the lower area is characterised
by year-round saturated or almost-saturated conditions, with a water table generally within 10 cm of the surface. Polygon rims
are still fairly wet, with the observed water table (shown by box plots) within 20 cm of the surface, while observations from
palsas have an unfrozen liquid fractional saturation of around 0.6 at 10-20 cm depth. Model outputs in Figure 8 will be
discussed in the following sections.

### 3.2.1 The importance of impeding baseflow (qbase)

Following the discussion on the treatment of baseflow in section 2.5.4 qbase configuration (modelling baseflow), model results
in Figure 8 are shown both with the TOPMODEL calculated qbase switched on in the top two panels and switched off in the
lower panels. This switch needs to be considered before examining the other aspects of hydrology because of the potential size
of its effect, the uncertainty in its implementation, and its departure from the standard hydrology of JULES. Switching off
qbase has a large effect on the palsa sites, but a small effect on the polygonal sites, where the thaw depth is shallow (Figure
8). We remind the reader that we have made the modification that qbase is always only permitted from saturated, unfrozen
layers. For the palsa sites and in standard JULES, impeding qbase is enough to significantly raise the water table, saturating
the soil at 20 cm depth for Iškoras. However, the soil only becomes fully saturated with the tiling scheme (for the low tile,
with qbase switched off). Conversely, if qbase is not impeded for these sites, the tiling scheme has negligible effect on the
saturation profile. Impedance of qbase is therefore a necessary condition for modelling palsa mire sites correctly in JULES.
This agrees with the simulations of Martin et al. (2019), who when varying the external water flux in their peat plateau model
found a "drainage effect" when transitioning from an external influx of +1.5 mm/day (+550 mm/year) to an outflow of -2
mm/day (-730 mm/year). This was enough to change the soil from saturated to well drained, decreasing the 1 m deep mean
annual temperature by up to $2°C$. With qbase on, qbase in the tiled scheme becomes over -200 mm/year on average for
Stordalen and over -500 mm/year on average for Iškoras (Figure B 3, Appendix B), which is at the drained end of the transition
found by Martin et al. We originally hypothesised that the calculated qbase could be too high for the low tile for palsa mires,
due to being calculated using the saturation of the wetter tile (section 2.5.4 qbase configuration (modelling baseflow)). While
qbase is significantly larger from the low tile than from standard JULES (~500 vs ~150 mm/year respectively for Iškoras),
qbase is still high enough to create a drained landscape even in standard JULES. From the sensitivity study, the individually
varying parameters runs showed that by varying $fd_2$ (the fractional scaling applied to the calculated qbase for the low tile) the
point at which saturation is reached can be seen (Figure C 1A, Appendix C). In fact, very little drainage is required for the
mires to become unsaturated, and as such there are very few parameter configurations from the Latin hypercube sampling runs
that achieve saturation for these sites. The sensitivity of the model to the calculated qbase points to the need for more accurate
determination of baseflow from permafrost wetlands. For the rest of this paper, the tiled configuration with qbase switched
off ('Full tiling (no qbase)') is regarded as the best performing model configuration and is the one presented in subsequent
figures.





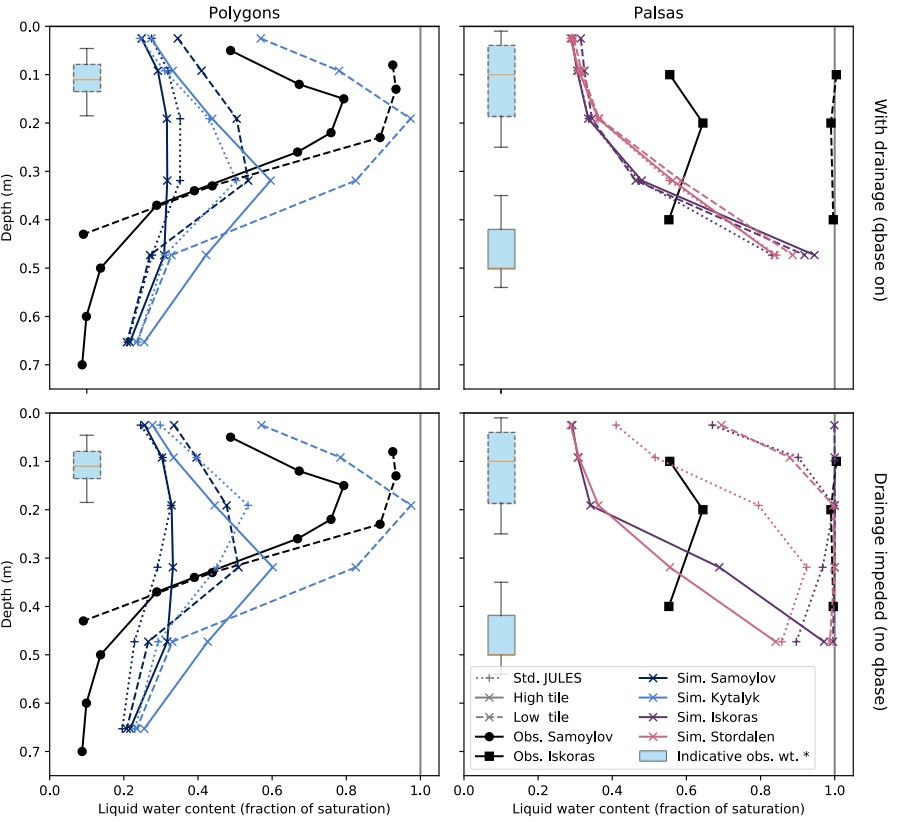

**Figure 8:** Average saturation profiles for July (2000-2016). Line colours show different sites, line styles differentiate high, low and standard JULES. *Indicative observed water table depths for polygons are based on average July observed water table depths for a polygon centre (Boike et al., 2019). Indicative observed water table depths for palsas are from measurements of 8 hollow points, not including ponded areas, and 5 palsa points in September 2007 (Klaminder et al., 2008). Water table depths for July from hollows in Iškoras were available but are not shown (C.T. Christiansen, H. Lee, unpublished data), as the range of values is contained within the range shown for Stordalen. The no qbase tiled runs represent the saturation profiles at palsa sites well, with water table depths within the range of observations. The same is true for the low tile for Kytalyk. Switching off qbase appears necessary for the low tile for palsa sites but has little effect on polygon sites. Lateral flows out of the palsa site high tile have a similar effect to that of the TOPMODEL calculated qbase. Observed soil moisture for the palsa at Iškoras at 0.4 m decreases vs that at 0.2 m due to still being partially frozen in July. Note that simulations show the average saturation for the layer whose midpoint is at the depth shown.

### 3.2.2 The effect of microtopography on soil moisture and fluxes

Compared with standard JULES with qbase off, the tiling scheme causes a splitting in the level of relative saturation (fraction of saturation) (Figure 8). For polygons, the raised tile remains broadly unchanged compared with standard JULES, whereas the lower tile gets wetter. For the palsas, the lower tile becomes slightly wetter and can now be fully saturated. The raised tile becomes much drier due to being able to laterally drain onto the lower tile, and the larger elevation difference of the palsa sites mean that the soil layers are less affected by the saturation of the low tile or even the level of the pond. Lateral drainage from the raised palsa tile has a very similar effect to that of qbase when switched on from the low tile or in the standard JULES simulation. In both cases, the lower tile is improved while the high tile remains, or becomes, too dry. Altogether, the palsa sites seem fairly well represented, though as we have seen, most of the credit for this is due to the disparity in drainage. While Kytalyk's centre is saturated, the polygonal sites remain too dry. Observations suggest high fractional saturations for rims of over 0.8 at around 0.15m depth, however simulated rims and even Samoylov's centre remain under 0.6. Nevertheless, in all cases the lower tile saturation is improved, which should have a positive effect on their respective soil temperatures and methane fluxes.



Figure 9 shows the average yearly simulated moisture fluxes for Stordalen. While fluxes do not indicate the end state of a system, comparing the magnitude and change in fluxes between standard JULES and the tiled configuration can give some indication of which fluxes are responsible for the change. Only Stordalen is shown as the other sites show a similar pattern. Compared to standard JULES, the high tile drying appears to be driven by lower snow mass causing decreased snowmelt water influx, as well as the addition of lateral egress out of the soil. Fluxes are balanced by decreased runoff due to the decreased

snowmelt. The low tile wetting appears to be driven by runoff decreasing almost to zero and the addition of run-on, an increase in snowmelt water influx due to increased snow mass, and the lateral egress of water from the high tile to the pond. The change in these fluxes is balanced by large amounts of pond evaporation. For Stordalen, there is also a small amount of lateral flow through the soil from the low tile to the high tile, though this is negligible for other sites. Changes in runoff and snowmelt water influx, due to spatial variability in snow depth as well as run-on and ponding, appear therefore to be the main drivers of

model soil moisture heterogeneity in this model, with lateral egress from the soil playing a smaller role at equilibrium. Unsaturated soil-to soil flows are negligible, due to a combination of horizontally adjacent soil layers being saturated and/or frozen. While lateral unsaturated fluxes may have greater relative importance if the landscape drains or becomes drier, the hydrological difference between tiles would be less pronounced in this case anyway. This suggests that simpler, water-table based models of lateral flow may be adequate in wetland environments.

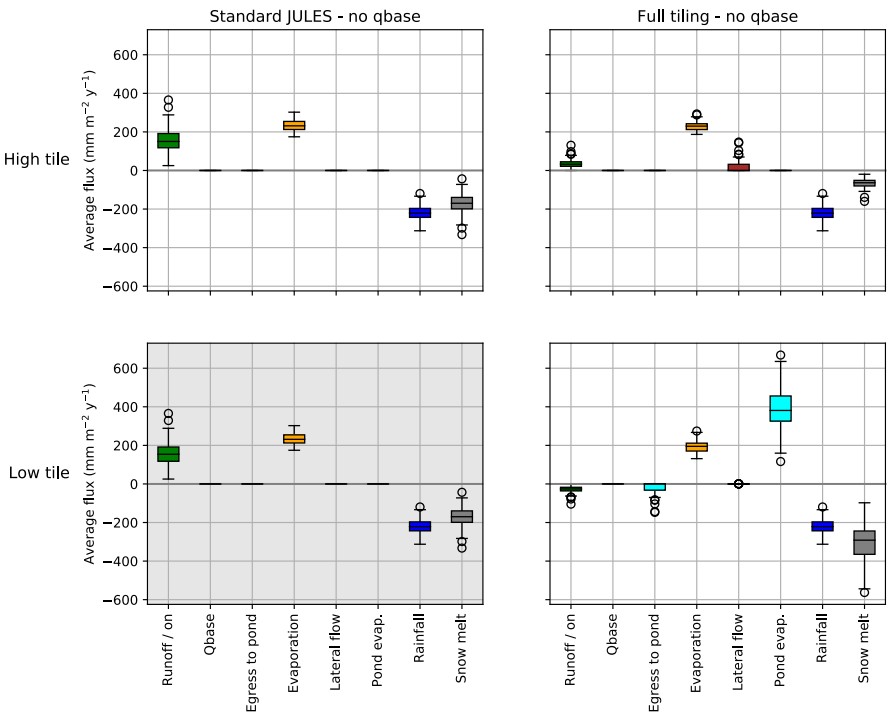

**Figure 9:** Average yearly simulated moisture fluxes for Stordalen for the period 1950 to 2016 (qbase off). High tile fluxes are shown top right, low tile fluxes bottom right. The plot for standard JULES is repeated (top and bottom left) to facilitate comparison with the tiled run. From the microtopography model, the change in snow melt and runoff have the greatest effect on soil moisture. Lateral flow is small, and in the form of egress from the soil of the high tile. The tiled run has a similar effect on the fluxes of the other sites.

**3.2.3 Distinguishing the impacts of model processes on soil moisture**

In order to directly attribute the effect of different model processes on soil relative saturation, a series of runs were performed with individual processes switched off in turn with tiled no qbase as the base configuration, the 'subtractive process switching'

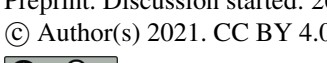



runs. The resulting effect on the relative saturation is shown in Figure 10. We find that although individually the effect of each process is small (aside from the effects of qbase, previously discussed, and lateral flow from the high tile in palsa mire sites,

dark blue in Figure 10) most processes have an effect in some conditions, and also that the importance of different fluxes varies with the degree of saturation. For example, if the mire is well drained or already fully saturated, ponding will have little effect.

For polygons, switching off snow redistribution has one of the largest effects, and reduces the low tile liquid water content as a fraction of saturation, henceforward referred to as fractional saturation, by $\sim -0.05$ to $-0.1$. Polygon hydrology is also

affected by the thermal state of the ground. For Samoylov, switching off lateral flows of water and heat (conductive and advective) has the largest effect, reducing the splitting in level of fractional saturation from 0.2 to 0.06. While this is the result of switching off two flows rather than one, the size of the effect is still surprising when considering that turning off lateral water flow alone has little effect and that the effect is larger than the reduction in splitting for switching off lateral heat alone. Indeed, this is the case even though the average lateral fluxes of water in the standard tiled run are negligible, and the

corresponding temperature splitting is very small (Figure B 3, Figure 11). Similarly, for Kytalyk, switching off lateral heat fluxes has the next largest effect after snow redistribution. This is perhaps unsurprising considering that this effect single-handedly reduces the temperature splitting from $2°C$ to almost nothing.  Switching off runoff and ponding have little effect. However, when the opposite method is tried and aspects of the model are switched on individually using std JULES as the base configuration (the 'additive process switching' runs), ponding and runoff can have a larger effect (Figure B 1, Appendix

B), particularly after snowmelt. This again indicates the differing relative importance of model processes at different degrees of saturation.

For palsas, as we have already seen, drainage (qbase and lateral flows alike) is of key importance. Switching off lateral flows of water allows the high tile fractional saturation to increase dramatically from under 0.4 to over 0.8 at 0.19m, while allowing

qbase from the mire results in the opposite effect and the mire draining, resulting in the fractional saturation at 0.19m of $< 0.38$ (see also Figure 8). For Stordalen, switching off lateral flows of heat, snow and runoff serve to slightly reduce the fractional saturation of the mire ($\sim -0.05$), but this effect is not seen for Iškoras.



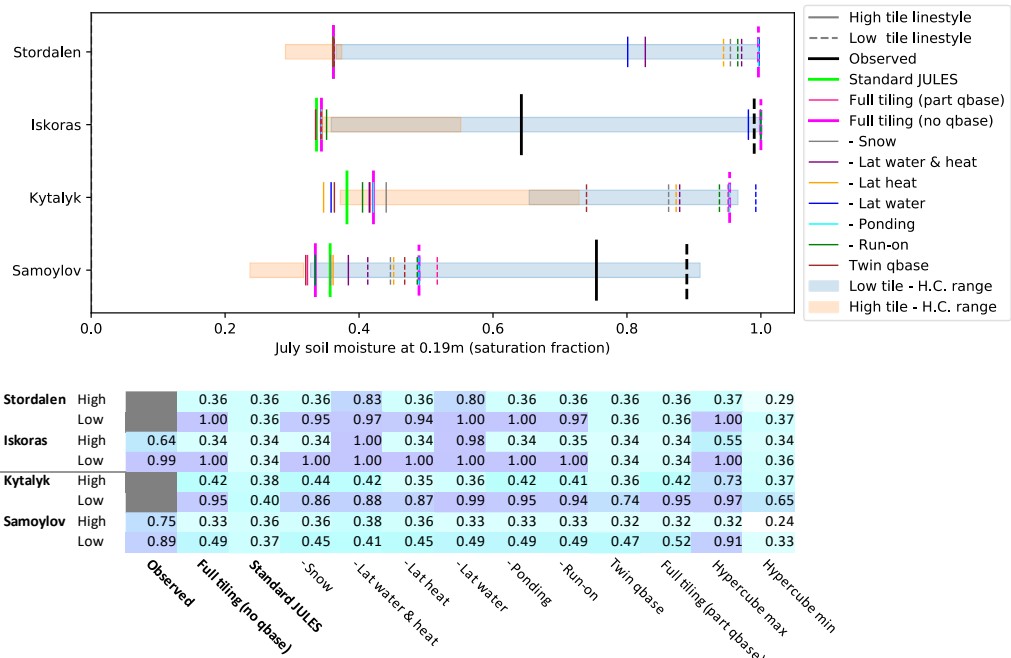

**Figure 10:** Average July soil moisture (2000-2016) at 0.19 m for the subtractive process switching runs, where model processes are switched off individually starting from the base configuration of Full tiling (no qbase) (thick pink). Solid lines show the high tile and dashed lines the low tile. Observations are shown in black, standard JULES in green. The range across the Latin hypercube sampling runs (H.C. range) are shown by the boxes. Model processes are shown to have a small effect individually. Lateral flows have a large effect on the high tile soil moisture at palsa sites. Snow and lateral flows of heat are important for polygon sites. Ponding appears unimportant in this figure, but can have a large effect depending on how close the soil is to saturation.

## 3.3 Heat and temperature

### 3.3.1 The effect of microtopography on soil temperatures

The two types of microtopography have different effects on soil temperatures. Figure 12 shows the observed and modelled soil temperatures for July at 0.19 m. The model summer temperature splitting is smaller than observed for palsas (3.5 vs 5°C), with the mean remaining lower and approximately unchanged compared with standard JULES, while polygons display small (0 to 0.5°C) temperature splitting, in agreement with observations. Thaw depths are likewise similar for high and low tiles at polygon sites, while palsa sites show a clear difference between palsa and mire (simulating permafrost and non-permafrost

conditions respectively). In the model, Iškoras shows signs of being on the edge of palsa degradation, with a talik forming for three consecutive winters after a warm summer in 2012 and greater winter snowfall for those years.

    Figure 11 displays climatologies for soil temperatures at 0.19 m depth, showing that the model is able to reproduce the overall behaviour for both types of landscape over the course of a year. Year round, the observed differences in temperatures between

the polygon rim and centre are comparatively small ($< 2.5°C$ at 20cm depth). However, polygon rims are observed to get colder in winter, be quicker to warm up in summer, can be warmer in summer, and cool down more quickly in winter. This general behaviour is reproduced in the tiled model, though the model lacks a substantial zero-curtain in the autumn when re-freezing. This is probably due to the modelled polygon sites being too dry. Only the low tile of Kytalyk approaches having a substantial zero-curtain behaviour, it also being the polygon tile nearest to the observed saturation. Microtopography tends to



split temperatures so that they fall either side of that of standard JULES. If the temperature modelled by standard JULES lies outside of the observed temperature splitting and is greater or smaller, the temperatures modelled by the tiled run also tend to be biased in the same direction and greater or smaller respectively. While polygon sites are close to observations in July, both standard JULES and the tiled configuration are generally colder than observed in summer, in addition to warming up and cooling down earlier. In addition to the aforementioned differences in soil moisture, part of the discrepancy in timing may be

due to the shorter period of forcing data snowfall. That Samoylov is too cold in winter may be in part due to having too little snow. Palsa mires show a more pronounced splitting in temperature for both the summer and the winter than polygons, and the model does well at capturing the difference in winter temperatures for Iškoras, and the mire zero curtain. However, the temperatures for both standard JULES and the tiled runs are too small in summer.

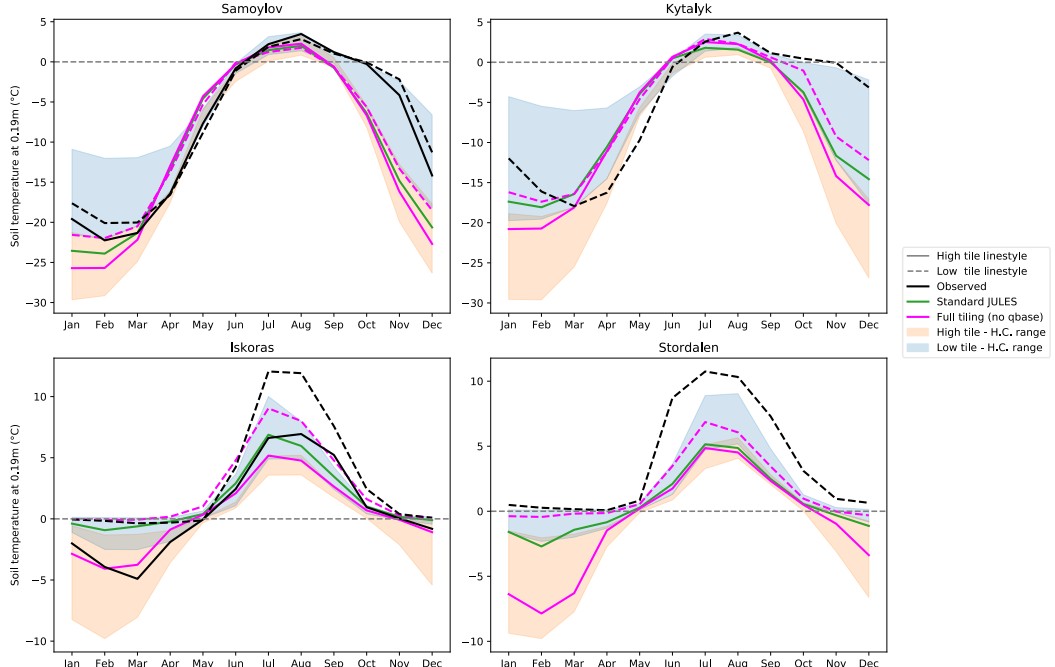

**Figure 11:** Climatologies (2000-2016) for soil temperature at 0.19m. Line styles show high and low tiles. Observations at this depth are not available for the polygon rim at Kytalyk and the palsa at Stordalen. H.C. = Latin hypercube sampling runs. The tiled model reproduces the seasonal temperature splitting behaviour, though lacks a substantial zero curtain for the polygon sites due to being too dry, and summer temperatures are cooler than observed for all sites. Soil temperatures for the tiled runs tend to lie either side of those for standard JULES.

**3.3.2 Distinguishing the impacts of model processes on soil temperatures and quantifying heat fluxes**

Figure 11 shows the effect on soil temperatures of the additive process switching runs, where model processes are switched on one at a time starting from the base configuration of standard JULES. Turning on the snow redistribution only results in almost the full winter temperature splitting for palsa sites, and too great a temperature splitting for polygon sites. The effect of snow redistribution on summer temperatures is smaller, though it is still responsible for the majority of the temperature

splitting. While their effect is smaller than that of snow, drainage, ponding and lateral flows of water increase the summer temperatures of the low tile, with the order of their respective magnitudes of effect approximately following that of their effects on soil moisture (Figure 12, Figure B 1 Appendix B). In particular we can see the difference between the best estimate and best estimate (no qbase) runs for the palsa mire sites, showing that the full temperature splitting cannot be achieved without a saturated low tile, though it remains to be seen if the high tile would also get significantly warmer if it too were wetter.





While its effect is not visible in Figure 11, lateral heat conduction has the effect of reducing the temperature splitting by -0.6 to -1.1°$C$ for palsa sites, and -1.1 to -2.0°$C$ for polygon sites (Figure B 2, Appendix B). This eliminates almost the entire temperature splitting for Kytalyk. There are almost no advective heat fluxes (Figure B 4, Appendix), however this may be due to the lack of saturated soil water fluxes in our model. The surface sensible and latent heat fluxes for the high and low tiles remain approximately the same as for standard Jules (Figure B 4, Appendix B). The modelled lateral heat conduction per unit

length of border for Iškoras is 2.9 Wm$^{-1}$. Over a year, the energy transferred per metre of border could melt 0.28 m$^3$ of ice. Interestingly, this is of the same order of magnitude as the 0.13 m$^3$ yr$^{-1}$ m$^{-1}$ of volumetric loss of permafrost peat plateau edge found by Martin et al., (2021). Of course, not all the lateral heat flux would go into melting ice, so this rate of thaw would be smaller. Nevertheless, it suggests the possibility that the modelled heat flux between just two tiles could be used to calculate the rate of lateral thaw at the interface, and hence the rate of areal change from one tile to the other.


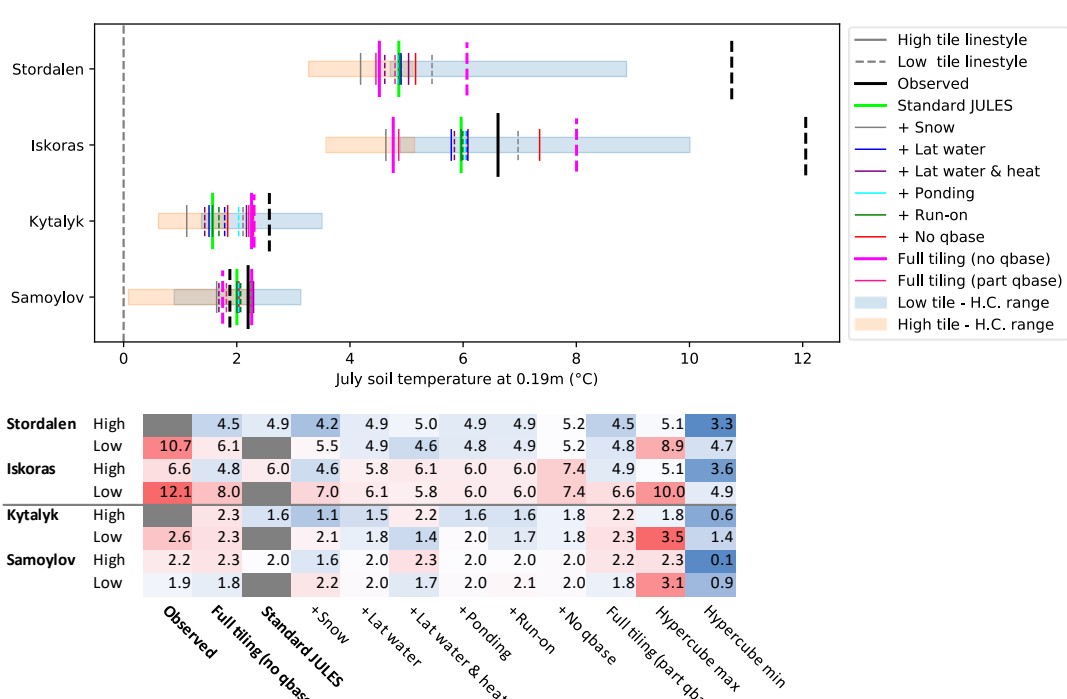

| | | Observed | Full tiling (no qbase) | Standard JULES | +Snow | +Lat water | +Lat water & heat | +Ponding | +Run-on | +No qbase | Full tiling (part qbase) | Hypercube max | Hypercube min |
|---|---|---|---|---|---|---|---|---|---|---|---|---|---|
| **Stordalen** | High | | 4.5 | 4.9 | 4.2 | 4.9 | 5.0 | 4.9 | 4.9 | 5.2 | 4.5 | 5.1 | 3.3 |
| | Low | 10.7 | 6.1 | | 5.5 | 4.9 | 4.6 | 4.8 | 4.9 | 5.2 | 4.8 | 8.9 | 4.7 |
| **Iskoras** | High | 6.6 | 4.8 | 6.0 | 4.6 | 5.8 | 6.1 | 6.0 | 6.0 | 7.4 | 4.9 | 5.1 | 3.6 |
| | Low | 12.1 | 8.0 | | 7.0 | 6.1 | 5.8 | 6.0 | 6.0 | 7.4 | 6.6 | 10.0 | 4.9 |
| **Kytalyk** | High | | 2.3 | 1.6 | 1.1 | 1.5 | 2.2 | 1.6 | 1.6 | 1.8 | 2.2 | 1.8 | 0.6 |
| | Low | 2.6 | 2.3 | | 2.1 | 1.8 | 1.4 | 2.0 | 1.7 | 1.8 | 2.3 | 3.5 | 1.4 |
| **Samoylov** | High | 2.2 | 2.3 | 2.0 | 1.6 | 2.0 | 2.3 | 2.0 | 2.0 | 2.0 | 2.2 | 2.3 | 0.1 |
| | Low | 1.9 | 1.8 | | 2.2 | 2.0 | 1.7 | 2.0 | 2.1 | 2.0 | 1.8 | 3.1 | 0.9 |

**Figure 12:** Average July soil temperature (2000-2016) at 0.19 m for the additive process switching runs, where model processes are switched on one at a time starting from the base configuration of standard JULES (light green). Solid lines
show the high tile and dashed lines the low tile. The range across the Latin hypercube sampling runs (H.C. range) are shown by the boxes. The model goes some way towards reproducing the observed temperature difference between tiles, though the modelled temperatures and temperature splitting are smaller than observed. Out of the model processes shown, snow redistribution and switching off qbase have the largest effect for palsa sites. For Kytalyk the effect of ponding and lateral flows can also be seen. The effect of lateral flows of heat cannot be seen in this figure, as it only has an effect where some
other factor is already causing a temperature difference.

**3.4 Methane**

Figure 13 shows methane fluxes calculated using the observed soil carbon profiles combined with soil temperatures from the 'standard JULES' and 'tiled (no qbase)' runs. In this figure, standard JULES has qbase switched on while tiled runs have qbase off. This is to represent the difference in methane fluxes between how JULES is usually configured and a fully
microtopographic approach. Based on the effect of the tiled model on temperatures seen in the previous section, we might expect to see a larger difference in methane production compared to standard JULES for the mire, and little difference with





respect to the polygon centres. Indeed, the total methane production for the polygons in JULES remains virtually unchanged for Samoylov and only 8 or 9% higher for Kytalyk. The palsa sites show an increase in methane fluxes vs standard JULES of 9 or 10% using the non-microbial scheme (NM), however when using the microbial scheme (M) for Iškoras this increases to

49%. This is likely a factor of whether or not the microbes can survive/grow in all soil layers – for example they are less likely to grow a population in permafrost soil layers. These results show that explicitly modelling microtopography can increase modelled methane fluxes vs the standard JULES in some cases. As previously mentioned, here we have used the observed, rather than modelled soil carbon profiles to calculate the methane flux. In reality, the carbon substrate might also be greater in the wet parts of the landscape compared to the landscape average (Wagner et al., 2003; Eckhardt et al., 2019), so the effect on

simulated methane that we show here may be amplified when soil carbon dynamics are modelled.

|  | Samoylov | Kytalyk | Iskoras | Stordalen |
|---|---|---|---|---|
| M Low vs std. JULES | 1.0 | 1.09 | 1.49 | 1.1 |
| M Low vs High | 0.98 | 1.09 | 1.67 | 1.24 |
| NM Low vs std. JULES | 1.01 | 1.08 | 1.1 | 1.09 |
| NM Low vs High | 1.0 | 1.1 | 1.21 | 1.2 |

**Figure 13:** Climatology of diagnosed methane fluxes calculated using modelled soil temperatures (2000-2016) and observed soil carbon profiles. Diagnosed methane fluxes for the high tile are shown for comparison only, as little methane would be emitted from these tiles due to the soil being drier. M = Microbial methane model, NM = Non-Microbial methane

model. Here, standard JULES has qbase switched on while tiled runs have qbase off. The tiled model has little effect on methane fluxes from polygons but increases the methane vs standard JULES for the palsa sites by 10 to 49% for the





microbial model. Simulated fluxes are lower than flux tower observations due to lower simulated temperatures. Methane chamber measurements for Stordalen are for 2002-2007 for the green season (days 119-288), and indicate the variability in methane fluxes due to hydrological conditions and thaw present at the site (Bäckstrand et al. 2010).


When comparing the simulated fluxes with flux tower observations (dotted lines in Figure 13), fluxes show reasonable agreement for palsa sites, though slightly lower than observed for Samoylov, but are more significantly lower than observed for Stordalen. This is what could be expected due to average modelled low tile summer temperatures for July and August being $< 1°C$ lower than observed for polygons, but $> 4°C$ lower for palsa sites (Figure 11). Previous modelling using a standard

version of JULES with the modification of a saturated soil column has been able to better reproduce the observed flux tower methane emissions for Stordalen (Chadburn et al., 2020, Figure 3). This underlines the primary importance of the core model setup in accurately modelling methane emissions, and that the difference made by modelling microtopography is only a refinement to this. In this case, the difference may be that the very upper layers of the lower tile in the model do not remain saturated year-round (Figure 8) unlike the fully saturated column in Chadburn et al., (2020). This drier surface layer of soil

provides an insulating effect, meaning that the soil does not heat up as much in summer. For the full tiling (no qbase) runs, Stordelen's pond is only seasonal, developing after snowmelt and disappearing before midsummer. In contrast, Iškoras has a persistent pond and a fully saturated lower tile, resulting in higher summer temperatures and methane fluxes more comparable to those observed for Stordalen.

While eddy covariance methane observations (EC obs.) have the benefit of providing a detailed timeseries and a landscape aggregate, they do not resolve small scale variations. For Figure 13, in order to compare the EC obs. with the modelled fluxes for the lower tile, for Kytalyk and Stordalen the EC obs. were selected for when the tower footprint was over the wetter area of the landscape, while for Samoylov the fluxes were divided by the observed wetland fraction (Table A 2, Appendix). Chamber measurements are able to resolve these spatial variations, however it is harder to get a landscape-scale average and

measurements often lack an extended timeseries. The average simulated fluxes are within the ranges of chamber flux measurements for Samoylov ($2.7 \pm 1.3\ mgC\ m^2 h^{-1}$, (Sachs et al., 2010)[1]) and for Kytalyk ($3 \pm 2 mgC\ m^2 h^{-1}$, (Huissteden et al., 2005)[2]) though these are not shown on Figure 13 due to the lack of a multi-year average. Methane chamber measurements from Bäckstrand et al. (2010) are shown for Stordalen. These measurements are the average from 2002-2007 for the 'green season' (days 119-288). Chamber measurements have been grouped as in (Johansson et al., 2006; Bäckstrand et al., 2010),

where the semiwet area (three chambers) corresponds to Sphagnum spp. and Carex spp. and the wet area (two chambers) to Eriophorum angsifolium. Three chambers were used for the palsa microsite. The wet area is completely saturated and permafrost free. The semiwet area has a water table fluctuating within 20 cm of the surface, and while this area not raised like the palsa, it is only partially thawed. A large difference in methane emissions can be seen between the wet and the semiwet areas. Our modelled lower tile fluxes are hard to directly compare with these areas. The lower tile sits in between the two in

terms of moisture and arguably in terms of temperature (based on thaw depths), and so the methane fluxes could be approximately correct for an area of the mire which is between wet and semiwet. However, as the saturated areas contribute a much larger share of the total methane emissions (see e.g. orange vs blue horizontal lines on Figure 13D), simulating methane emissions from unsaturated parts of the landscape will have little impact on the upscaled fluxes.

Another problem for upscaling is that the method of calculating methane emissions used here is now decoupled from changes in the water table. In reality, the thickness of the oxic layer determines the amount of substrate unavailable to methanogens

[1] Three low-centred polygon centres of three chambers each, uncertainty shows standard error in spatial variability. Seasonal average (10th July to 18th September 2006). This study confirmed that the chamber measurements matched up to those measured by the eddy covariance tower using the wetland fraction used here.
[2] Two low-centred polygon centres (27th – 30th July, 2004). Large variability can be seen for fluxes from the surrounding area.





and affects how much of the generated methane is oxidised. This is represented in standard JULES by varying the wetland fraction depending on the water table level, as described in the introduction. Here, the low tile area is fixed, and so our calculation of the fluxes will increasingly be an overestimate as the water level drops below the surface. Future versions of the

model will therefore require changes in the water table and the methane fluxes to be linked. As Chadburn et al. (2020) observe, doing so would enable the microbial dynamics to be better modelled, as methanogens would be unable to survive in layers where the saturation is varying rapidly. Adding water table dynamics would also add to the feedback on the methane fluxes described previously of an insulating dry layer limiting summer soil temperatures, making the low tile emissions more sensitive to changes in the water table. Again, this is a contributing factor to the observed spatial variability in methane emissions and

must be considered when upscaling.

### 3.5 Sensitivity study

The sensitivity study investigated the response of the model to varying model parameters using the methods described in section 2.5.2 Configurations of JULES simulations. Using the estimated possible ranges of parameter values, we determined the range of possible modelled soil temperatures for each site and quantified how the uncertainty in model parameters affects

the uncertainty in model output. These tests are important as they confirm that the model will not give unreasonable results if the parameters are set differently, while also indicating which parameters are in most need of constraining.

Figure 11 and Figure 12 show the variability in the temperature splitting (TS, the difference between high and low tile temperatures) for July at 0.19 m as a result of varying parameters using Latin Hypercube Sampling (LHS). It can be seen that

this variability is comparable to the size of the observed TS itself. For the polygonal sites, the choice in parameters can result in the TS being larger by up to $\sim 3°C$, as the usual splitting is small ($0.5°C$ for Samoylov). A similar increase from the usual modelled splitting is also possible for the palsa sites, however as the modelled TS is already smaller than observed, the resulting splitting can only be up to $\sim 1.5°C$ larger than observed. The parameter ranges used for the LHS are generous, and so we can be confident that including microtopography is 'safe' and will not result in unreasonable modelled temperatures. However,

that the variability in the TS is similar to the size of the modelled TS means that care must be taken when choosing parameter values.

Table 4 gives approximate estimates of parameter uncertainties by site, which were then used alongside the results from the individually varying parameters runs (as in Figure C 1C, Appendix) to find the resulting effects of the parameter uncertainties

on the size of the July TS at 0.19 m depth. The parameter uncertainties are estimates of our uncertainty in the parameters used, and not a quantification of site level variability based on any large-scale survey. For Kytalyk, our choice of parameters appears to have resulted in a local minimum for the TS, such that for four of the parameters, variation in either direction results in an increase of magnitude of the TS. As a result, either tile could become the warmer tile, so microtopography could either increase or decrease the fluxes. The largest resulting change in the TS as a result of the uncertainty in an individual variable is $1.5°C$

for the palsa sites and $0.7°C$ for the polygon sites. This can be a large fraction of the overall TS and is larger than the effect of many of the individual model processes, reinforcing the need to carefully chose model parameters. This is particularly the case for $A_1$ (the high tile area, which affects the depth of snow on the low tile), $fd_1$ (fraction of drainage from the low tile, which has been previously discussed), and $\Delta x$ (the horizontal distance, which is hard to measure but effects the magnitude of lateral fluxes). At the other end of the scale, uncertainties in the values of $fpd$ (maximum pond depth), $fd_1$ (drainage from high tile),

and $fro$ (run-on fraction) have little individual effect ($\leq 0.1°C$) on the TS. There can be a large variability in palsa heights at an individual site, for instance, Olefeldt and Roulet (2012) record a range of 0.5 to 2 m for Stordalen. It had therefore been a concern that a single pair of tiles may not be representative if this also corresponds to a large variability in temperature. However, Figure C 1C, (Appendix) shows that there is little change in the modelled temperatures once the palsa elevation is



above 0.5m, meaning that having only two tiles can be a valid representation of multiple palsas whose heights are above this
threshold. While we have taken care in choosing model parameters based on observations, it is recommended parameter validation be implemented for future users, so that unreasonable or even unphysical combinations of parameters are not allowed. The sensitivity study also provides further motivation for a dynamic representation of the parameter $\Delta x$, as this currently responsible for a $0.6°C$ uncertainty in the TS.

**Table 4:** Showing estimated parameter values (Value) and uncertainties (+/-) by site, and the resulting effect of the parameter uncertainty on the size of the July temperature splitting at 0.19 m depth (Effect). </> indicates whether increasing the parameter causes the temperature splitting to increase (1), decrease (-1) or the presence of a local maxima (0 +) or minima (0 -). Parameters with the largest effect

| | Samoylov | | | | Kytalyk | | | | Iskoras | | | | Stordalen | | | |
|---|---|---|---|---|---|---|---|---|---|---|---|---|---|---|---|---|
| Parameter | Value | +/- | Effect (°C) | </> | Value | +/- | Effect (°C) | </> | Value | +/- | Effect (°C) | </> | Value | +/- | Effect (°C) | </> |
| $\Delta z$ (m) | 0.38 | 0.15 | 0.1 | 1 | 0.35 | 0.15 | 0.05 | 1 | 0.68 | 0.5 | 0.5 | 1 | 2 | 1 | 0.5 | 1 |
| $A_1$ (m²) | 70 | 40 | 0.25 | -1 | 23.4 | 15 | 0.4 | 0 - | 25 | 50 | 0.4 | 0 + | 25 | 50 | 1.25 | 1 |
| $A_2$ (m²) | 58 | 30 | 0.1 | -1 | 19.6 | 15 | 0.5 | 0 - | 25 | 50 | 0.5 | 0 + | 25 | 50 | 0.6 | -1 |
| $\Delta l$ (m) | 26.7 | 20 | 0.35 | 1 | 15.7 | 15 | 0.7 | 0 - | 5 | 5 | 0.5 | -1 | 5 | 5 | 0.3 | -1 |
| $\Delta x$ (m) | 2.1 | 1.5 | 0.65 | -1 | 1.2 | 0.6 | 0.5 | 0 - | 2 | 1.5 | 0.6 | 1 | 2 | 1.5 | 0.5 | 1 |
| $sc$ (m) | 0.05 | 0.03 | 0.15 | 1 | 0.05 | 0.03 | 0.3 | -1 | 0.05 | 0.03 | 0.15 | -1 | 0.05 | 0.03 | 0.3 | -1 |
| $fpd$ | 0.9 | 0.2 | 0 | 1 | 0.9 | 0.2 | 0.1 | 1 | 0.4 | 0.3 | 0.1 | 1 | 0.4 | 0.3 | 0 | 1 |
| $fd_1$ | 0 | 1 | 0.05 | -1 | 0 | 1 | 0 | -- | 0 | 1 | 0 | -- | 0 | 1 | 0 | -- |
| $fd_2$ | 0 | 1 | 0.05 | 1 | 0 | 1 | 0.05 | 1 | 0 | 1 | 1.5 | -1 | 0 | 1 | 1.5 | -1 |
| $fro$ | 0.9 | 0.3 | 0 | -- | 0.9 | 0.3 | 0.07 | -1 | 0.8 | 0.5 | 0 | -- | 0.8 | 0.3 | 0.05 | 1 |

The results of the individually varying parameters runs also give insight into the effect of varying parameters on winter soil temperatures, and particularly the model response to parameters affecting the snow scheme. While the main behaviours follow directly from the model equations (1 & (2), it is worth noting that soil temperatures are most affected by changes in snow depth when snow depths are shallower. For instance, soil temperatures are highly sensitive to the snow catch parameter ($s_c$) for the high tile, where the sensitivity tests show that a difference of 5 cm can result in as much as $2°C$ difference at 0.19 m (Figure C
1.B, Appendix). However, the same is not true of the response of the low tile temperatures to snow catch as the snow is deeper, so changes have less effect. Similarly, changes to $A_1$ (the area of box 1) are a key control on snow depths on the low tile and can affect the snow depth by a much greater amount but effect the temperature less. For instance, changing the $A_1$ by 50 m² for Stordalen results in changing the low tile snow depth by around 0.5 m and a temperature increase of around $1°C$.

**4 Summary and outlook**

In this study we have constructed and tested an explicit representation of microtopography within the JULES land surface model. We have followed the two-tile approach suggested by Aas et al., (2019) and Nitzbon et al., (2019), though in order to implement such a model in JULES we have taken a different approach to implementing lateral fluxes, runoff and ponding. Our purpose has been to test the efficacy of an explicit representation of microtopography further: by comparing the model output with observations of snow depth, soil moisture and temperature at additional sites, though the study of the model's behaviour
and uncertainty though testing a range of parameter values, and through gauging the impact of modelling microtopography on modelled methane fluxes. We have shown that a simple two-tile representation of microtopography (Figure 2) is able to improve modelled snow depths (Figure 7), soil moistures (Figure 8) and temperatures (Figure 11) in JULES for four sites, and that this can lead to an increase in the modelled methane fluxes in some cases. These changes would affect the results of modelling the northern high latitudes in their current state, though have implications for and will be affected by future





landscape changes due to ground ice thaw and change in wetland area. The increase in methane fluxes was appreciable for
      only the palsa-mire sites (+10 to 49%, Figure 13) due to the small difference in temperatures between tiles at the polygon sites
      (Figure 11). We hypothesise that a similar effect on methane fluxes would be seen for other wetlands in the discontinuous
      permafrost region. This is because the model processes are general to all forms of microtopography, and because through this
      study we have found a weak dependence of soil temperatures on elevation differences greater than 0.5 m (Figure C 1C,
Appendix). To test this hypothesis would require data from additional sites, so this study underlines the need for more
      microtopographically resolved data to be available.

      Snow depth was a key control on the modelled soil temperatures of each tile (Figure 12). Despite its simplicity, the snow
      scheme performed well at the modelled sites. While these sites are representative of widespread landforms, it remains to be
seen if the scheme would perform as well in sites where the microtopography does not show such clear small-scale repetition.
      Investigating the impact of linking the snow catch parameter to the vegetation height is a key next step and will be important
      for modelling snow-vegetation feedbacks. The impact of microtopography on modelled plant communities was not examined
      in this study as work needs to be undertaken to implement plant functional types in JULES which are more representative of
      arctic plants and their variety, and which have the correct response to saturated soils. The ability to model different plant
functional types in microtopographically distinct areas would however be interesting with respect to peat formation and the
      effect of different plant functional types on methane emissions (Cooper et al., 2017; Rupp et al., 2019). Combining this model
      with the ability to model peat formation as in Chadburn et al. (submitted) could lead to interesting dynamics as the tile
      elevations change, and would enable the future of features such as drained thermokarst lake basins to be modelled.

This study reiterates that making wetlands wet is a clear requirement for accurate representations of soil temperatures and
      therefore methane emissions from high-latitude wetlands (Figure 12). An explicitly modelled wetland 'tile', even if purely to
      provide more accurate soil temperatures for calculation of methane emissions, could therefore be a worthwhile step forward
      from models where gridcell soil temperatures are spatially homogeneous. Differences in the snowmelt, lateral drainage and
      runoff between tiles, as well as ponding on the low tile, were instrumental in modelling the effects of microtopography on
hydrology. As such, it is difficult to suggest any of these processes that could be readily eliminated. However, several aspects
      of the hydrology were identified which require consideration for future modelling. While much of our effort was directed at
      modelling unsaturated flows, it was found that these fluxes played only a small part in modelling these wetland environments.
      Where lateral fluxes were appreciable, they were usually from a saturated layer to an unsaturated one. Modelling the lateral
      fluxes based on the position of the water table may therefore provide an acceptable basis for a simplified implementation of
these flows, and one which may better model advective flows of heat. The amount of baseflow modelled from the low tile has
      a large effect for the palsa mires. However, the current implementation of baseflow in JULES is not very suitable for a
      microtopographic model and also does not represent water influx, which is important for some wetlands. Consideration
      therefore needs to be given as to how the landscape-scale lateral flow out of and into wetlands could best be modelled. To
      allow for changes in the soil wetness, the methane flux calculation will need to be linked to the water table depth, which will
also enable methanogen microbial dynamics to be better modelled. This will increase the sensitivity of the methane flux to the
      water table depth that is already present due to the effect of soil moisture on soil temperatures. This sensitivity is part of the
      reason for the observed spatial variability of methane emissions from wetter areas. There is therefore a question of how
      upscaling the methane fluxes from those calculated for the low tile can be done in a representative way. Bechtold et al., (2019)
      suggest that for peatlands the TOPMODEL formulation of calculating the distribution of the water table should be replaced
with a normal distribution based on the microtopographic variability, alongside other modifications to the hydrology. This
      distribution could then be used alongside temperatures of tiles of different wetness to better represent the emissions from the
      whole of the mire. An alternative would be to explicitly represent the hydrological gradient using a larger number of tiles



and/or a representation of hillslopes (Nitzbon et al., 2020a; Swenson et al., 2019; Hazenberg et al., 2015). Both of these approaches could also provide a route to better modelling the landscape or catchment-scale baseflow. Currently the model does not include a treatment of surface waterbodies beyond being a simple surface store, for which an additional pond or lake tile would be beneficial, as in Langer et al., (2016). Using a greater number of tiles may also be useful for the modelling of more complex features, for example by using three tiles to represent the centre, rim and trough in a polygonal landscape (Nitzbon et al., 2019), or areas of land, wetland and lake (Schneider von Deimling et al., 2015). Indeed, Nitzbon et al. (2020) take this further, connecting multiple polygon units together to represent landscape-scale drainage and mesotopography.

Modelling microtopography provides a foundation for representations of thermokarst in Earth system models, and for answering key scientific questions about the impact of abrupt thaw processes on emissions and the future extent of wetlands. Scaling to a pan-arctic model requires consideration of how model parameters such as tile areas and elevations can be assigned on this scale, and a representation of thermokarst requires a model of how these parameters can change over time. This study provides motivation for it being possible to assign parameters that are representative at scale, but also a measure of the uncertainty as a result of estimating these parameters. In particular, we find that within the expected ranges at these sites, most parameters result in an associated uncertainty in July soil temperatures at 20 cm depth of under $0.8°C$, while the effect of the uncertainty in the high tile area and in the fractional drainage from the low tile result in an uncertainty on these temperatures of under $1.6°C$ (Table 4). We also find a relatively weak dependence of summer soil temperatures on elevation difference within the expected ranges at each type of site. It would therefore be reasonable to associate a given elevation difference with a particular landform. Previously, thaw models based on the thawing of excess ice and subsequent vertical elevation change have been used to simulate the future evolution of polygonal landscapes and palsas / peat plateaus (Langer et al., 2016; Aas et al., 2016, 2019; Nitzbon et al., 2019, 2020a; Cai et al., 2020; Martin et al., 2021). These approaches are particularly suited to polygonal landforms where the relative elevations of tiles determine the transformation between low centred and high centred polygons, affecting the hydrology of these landscapes (Nitzbon et al., 2019, 2020a, b). However, for areas of peat plateau, and for the expansion of thermokarst lakes, it may be more suitable to consider an area-based lateral thaw approach as this may require fewer tiles. We note that the modelled lateral heat conduction between tiles for Iškoras corresponds to melting 0.28 m$^3$ of ice per year per metre of palsa-mire border, which is of the same order of magnitude as the 0.13 m$^3$ yr$^{-1}$ m$^{-1}$ of volumetric loss of permafrost peat plateau edge found by Martin et al., (2021), indicating that such an approach may work. Recently, remote sensing and machine learning tools have enabled the mapping of the location and areas of different permafrost landforms for large regions, and the mapping of how these are changing (Nitze et al., 2018). At the same time, climate envelope models have allowed for pan-arctic assessment of climatic suitability for palsas (Fewster et al., 2020) and ice wedge polygons, and which areas may be subject to change in the future. These data will enable microtopographic models to be initialised and provide a basis for model evaluation. The time is therefore ripe for explicit representations of microtopography to be set up and assessed on a pan-Arctic scale.

## 5  Code and data availability

Model code and Rose suite configuration files are freely available from the Met Office Science Repository Service ( https://code.metoffice.gov.uk/) upon registration and completion of a software licence. Details on accessing and running JULES configurations can be found in Wiltshire et al. (2020). The tiled model code (vn5.4_microtopography) can be found at: https://code.metoffice.gov.uk/svn/jules/main/branches/dev/noahsmith/r17292_vn5.4_lateral_9box (revision 20192), which is itself a branch from a modified version of JULES 5.4 (https://code.metoffice.gov.uk/svn/jules/main/branches/dev/sarahchadburn/vn5.4_microbial_ch4 , revision 15781). Tiled model runs use Rose suite u-bo877x (https://code.metoffice.gov.uk/svn/roses-u/b/o/8/7/7/) which was built off of the base



configuration u-an231 ('standard JULES') (https://code.metoffice.gov.uk/svn/roses-u/a/n/2/3/1/, revision 175882). Further

configuration   files   for   sensitivity   suites   can   be   found   at
https://github.com/noahdsmith/JULES_microtopography_model_rose_suites/ (last access: 7[th] August 2021, Smith 2021).
Model output, processed observational data and plotting code is available at https://doi.org/10.5281/zenodo.5565163 (last
access: 13[th] October 2021). Sources for site observations can be found in Table A 2 (Appendix).

**Author contribution**

NDS developed the model, performed simulations and wrote the first version of the manuscript. EJB prepared forcing data
and set up the JULES suite u-an231. SEC, EJB, RT and HR contributed to model development. SEC, JB, SW, HL, CTC, BE,
TF, KSA, IHJA provided model forcing and evaluation data. All authors contributed to the text of the manuscript.

**Competing interests**

The authors declare no competing interests.

**Acknowledgements**

In  addition to those mentioned in Table A 2, we thank Han Dolman from the department of geosciences, VU University
Amsterdam for contributing methane and soil temperature data for Kytalyk.



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





**Appendices**

**Appendix A – Site information and data sources**

**Table A 1** Further site information. Acronyms - MAAT: Mean Annual Air Temperature, MAP: Mean annual precipitation, WM: Warmest month, CM: Coolest month. Other than 'Recent MAAT' climatic variables are aggregated for period given by ''variable period'.


| Site | Samoylov | Kytalyk | Iškoras | Stordalen |
|---|---|---|---|---|
| Location | Lena River Delta, Siberia 72.22°N, 126.467°E | Sakha, Siberia 70.83°N, 147.485°E | Northern Norway 69.3003°N, 25.3460°E | Abisko, Northern Sweden 68.35°N, 18.817°E |
| Type | Polygonal tundra | Polygonal tundra | Palsa mire | Palsa mire |
| Recent MAAT | -11.6°C 2006 - 2011 | -12.2°C 2000 - 2019 | -0.8°C 2006 - 2019 | 0.6°C 1995-2007 |
| MAAT | -12.5°C | -13.6°C | -2.1°C | -0.6°C |
| CM MAT | -30.3°C January | -34.1°C January | -15.2°C January | -10.8°C January |
| WM MAT | 10.1°C July | 9.7°C July | 13.2°C July | 11.7°C July |
| MAP | 125 mm | 202 mm | 339 mm | 305 mm |
| Variable period | 1998 – 2011 | 1940 – 2019 | 1876-1980 temp 1895-1990 precip | 1913-2007 |
| Source | (Boike et al., 2013) | Chokurdakh Station 21946 (World Meterological Association - Global Historical Climatology Network, 2021) | Karasjok Station 1065 (World Meterological Association - Global Historical Climatology Network, 2021) | (Callaghan et al., 2010) |
| Thaw depth | 0.41 to 0.57 m | 0.2 to 0.3 m dry 0.4 to 0.5 m wet | 0.4 to 0.65 m for stable permafrost | 0.5 m palsas 1 to 3 m mire |
| Organic layer thickness | 0 to 0.15 m dry < 0.2 m wet | 0.1 to 0.15 m wet | 1.5m | 0.5 to > 1 m peat on Palsa |
| Permafrost thickness | 400 to 600 m | Not available | Not available | 10 to 20 m |
| Dry area vegetation | Moss Hylocomium splendens, dwarf shrub Dryas punctata | Dwarf shrubs Bela nana and Salix pulchra (diamondleaf willow), sedge Eriophorum vaginatum (cottongrass), moss and lichen | Lichen, moss, Ericaceae (heather) e.g. Empetrum nigrum and Rhododendron groenlandicum, Rubus chamaemorus (cloudberry) | Dwarf shubs e.g. Empetrum hermaphorditum, sedges Eriphorum vaginatum, mosses Sphagnum fuscum and Dicranum elongatum, lichens Cetraria spp. and Caladonia spp. |
| Wet area vegetation | Mosses Drepanocladus revolvens and Meesia | Moss Sphagnum, Potentilla palustris, sedges Carex | Moss sphagnum, sedges Eriophorum (cottongrass), shrubs Betula (birch) | Moss Sphagnum balticum and sedges (Carex spp., Eriophorum spp.) |





| | triquetra, sedge Carex chordorrhiza | | | |
|---|---|---|---|---|
| Further details | (Boike et al., 2019) | (Parmentier et al., 2011; van der Molen et al., 2007) | (Kjellman et al., 2018; Martin et al., 2019) | (Olefeldt and Roulet, 2012; Jammet et al., 2015, 2017; Klaminder et al., 2008) |

**Figure A 1**: Date ranges and depths for available soil observations. When comparing with the JULES soil level at 0.191m depth, the closest soil observation was used. No equivalent level was available for this depth for the Palsa at Stordalen.

**Table A 2:** Sources of site observations.

| | Samoylov | Kytalyk | Iškoras | Stordalen |
|---|---|---|---|---|
| Soil temperatures and moistures | (Boike et al., 2019) | (van der Molen et al., 2007) | (C.T. Christiasen, H. Lee, unpublished data) | Mire temperatures (Jammet et al., 2015, 2017) Palsa temperature from Storflaket mire, near to Stordalen (Johansson et al., 2011; Åkerman and Johansson, 2008) |
| Eddy covariance methane flux | 2012 - 2017 (Boike et al., 2019; Sachs et al., 2010) also on fluxnet: | 2012, 2015 & 2016 (Parmentier et al., 2011) | -- | 2012-2014 (Jammet et al., 2017) also on  fluxnet: http://sites.fluxdata.org/SE-St1/ |





| | http://sites.fluxdata.org/RU-SAM/ | | | |
|---|---|---|---|---|
| Wetland fraction* | 23% (uses wet tundra, overgrown water but not open water) (Muster et al., 2012) | Wind from 250 to 330° (Parmentier et al., 2011) | -- | Wind from 210 to 330° (Jammet et al., 2015) |
| Soil carbon | (Chadburn et al., 2017) | (Chadburn et al., 2017) | (Kjellman et al., 2018; Sannel and Kjellman, 2018) | (Jammet et al., 2015) |
| Snow | Survey across multiple polygons for rims & centres April 2013 (Gouttevin et al., 2018) Polygon centre timeseries 2012 - 2019 (Boike et al., 2019) | -- | Timeseries at different elevations 2018 - 2020 (S.E. Chadburn, unpublished data) | -- |
| Other | | | Air temperature (B. Etzelmüller, K. S. Aas, unpublished data) | |

\* Used for scaling eddy covariance data to flux m$^{-2}$, from (Chadburn et al., 2020)

**Appendix B - Additional figures**

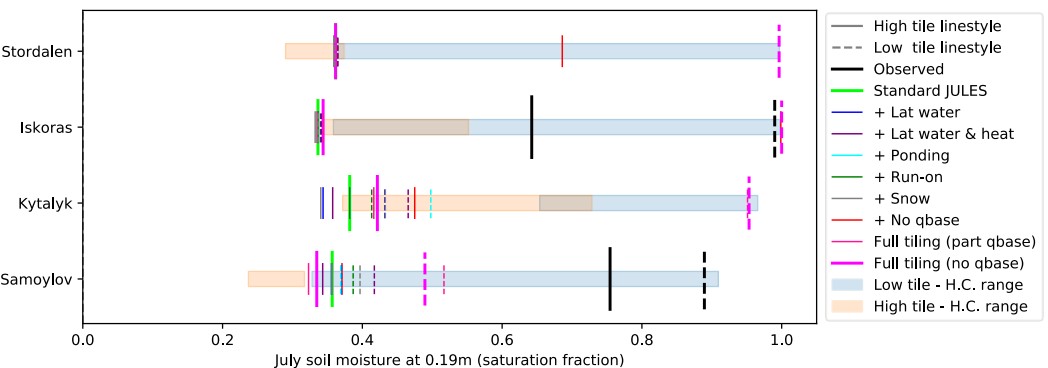

**Figure B 1:** Average July soil moisture (2000-2016) at 0.19 m for the additive process switching runs, where model processes are switched on one at a time starting from the base configuration of standard JULES (light green). Solid lines show the high tile and dashed lines the low tile. The range across the Latin hypercube sampling runs (H.C. range) are shown by the boxes. Processes tend to have little effect individually, however for the palsa sites it can be seen that ponding, run-on and redistribution of snow act to make the low tile wetter.





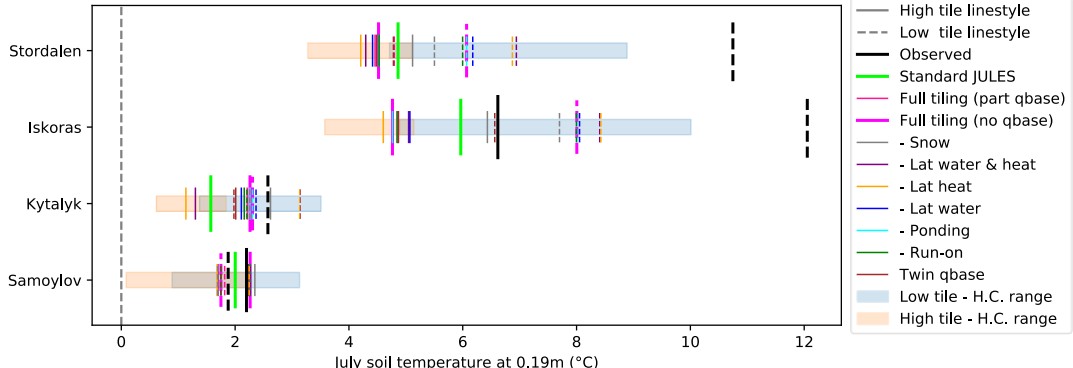


**Figure B 2:** Average July soil temperature (2000-2016) at 0.19 m for the subtractive process switching runs, where model processes are switched off individually starting from the base configuration of Full tiling (no qbase) (thick pink). Solid lines show the high tile and dashed lines the low tile. The range across the Latin hypercube sampling runs (H.C. range) are shown by the boxes. The effect of lateral heat fluxes on reducing the temperature splitting can be seen by comparing the 'Full tiling

(no qbase)' run with the '- Lat heat' run, particularly for Kytalyk.

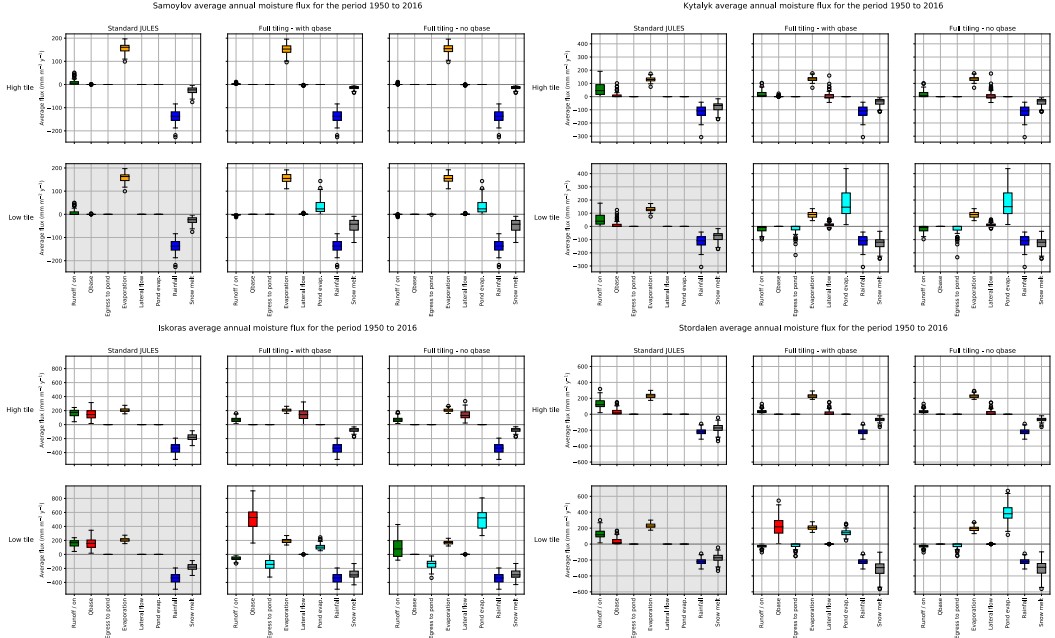

**Figure B 3:** Average yearly simulated moisture fluxes for the period 1950 to 2016 for all sites. The plot for standard JULES is repeated to facilitate comparison with the tiled run. From the microtopography model, the change in snow melt and runoff have the greatest effect on soil moisture. Lateral flow is small, and in the form of egress from the soil of the high tile.





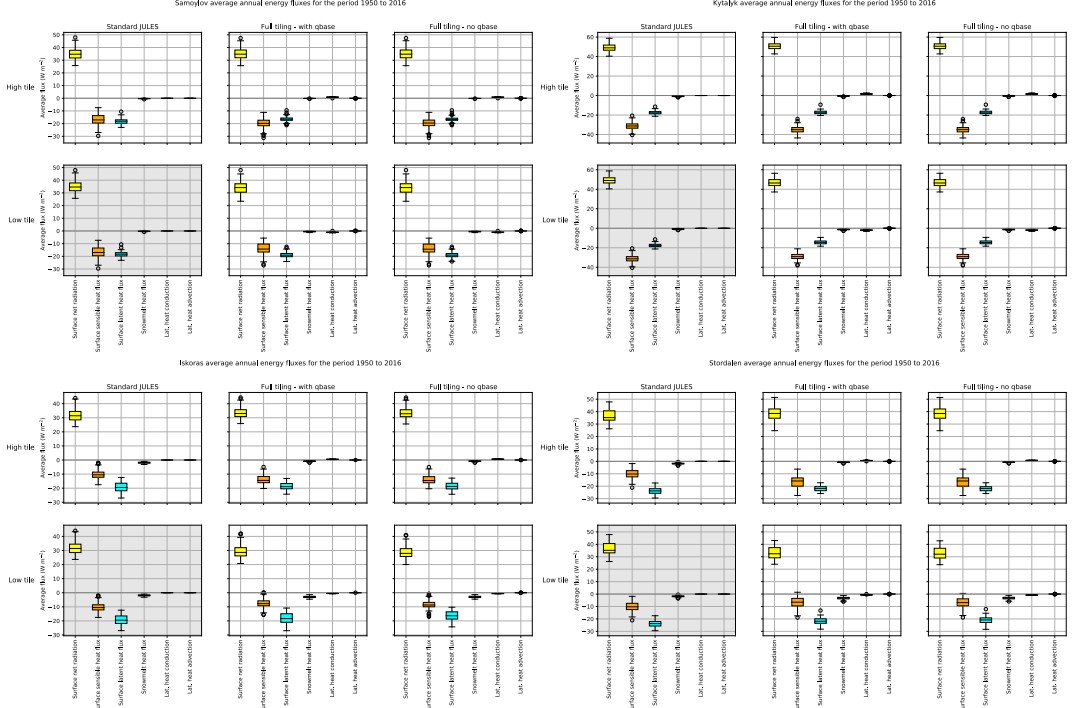

**Figure B 4:** Average yearly simulated energy fluxes for the period 1950 to 2016 for all sites. The plot for standard JULES is repeated to facilitate comparison with the tiled run. Surface sensible and latent heat fluxes remain approximately the same for the two tiles in the tiled runs and the single tile in the standard JULES run.



**Appendix C – Sensitivity study**

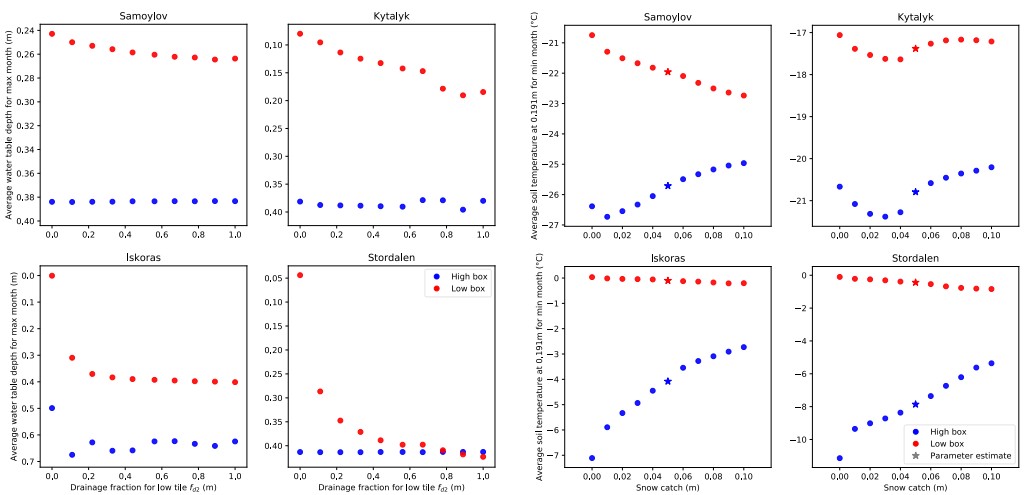

**A.** Drainage as a fraction of that calculated by TOPMODEL for the lower tile, $f_{d2}$, vs maximum water table depth. Little drainage is required for the mires at the palsa sites (Stordalen and Iskoras) to become unsaturated. Polygon sites are less affected.

**B.** Snow catch vs winter soil temperature at 0.19 m. High box temperatures are most sensitive to this parameter.

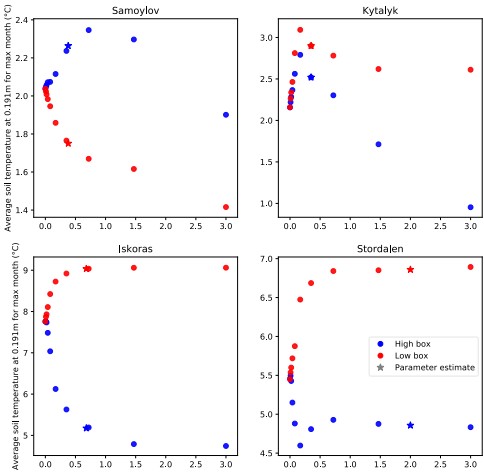

**C.** Elevation vs summer soil temperature at 0.19 m. For the palsa sites, there is a weak dependence of temperature on elevation for elevations above 0.5 m.

**Figure C 1:** Selected results from the varying individual parameter runs, showing an output variable against the parameter being varied. Red and blue show the high and low tiles respectively. Panes B and C have qbase off for both tiles.
