# Peer review of "Explicitly modelling microtopography in permafrost landscapes in a land-surface model (JULES vn5.4\_microtopography)"

_Geoscientific Model Development, 2021_

## Author Response (AR1)

**Author response to comments**
(in brown)

**Anonymous Referee #1**

Smith et al. present a study in which: 1) JULES, the land component of the UK Earth System Model, is updated to represent microtopographic relief in permafrost settings using a two-tile approach, allowing for redistribution of snow, soil moisture, heat between tiles, and surface water accumulation in the lower tile; and 2) the new model is validated against observations from two ice wedge polygon field sites in Siberia and two palsa field sites in Scandinavia. Similar approaches have been used in the past to study these same land forms in other land surface models, but this study is the first to implement these changes in JULES. The authors clearly define the scope of their study and its limitations, while suggesting next steps to build on their approach in the future. In its current iteration, the implementation of microtopography in JULES allows for reasonable snow redistribution and temperature splitting between high and low tiles. It also allows for the simulation of perennially saturated soils in the lower tile of one of the palsa sites, which meaningfully impacts simulated methane emissions.

Overall the paper is a meaningful step forward in representing permafrost environments within Earth system models and worthy of publication in GMD. However, I recommend that the authors address the following concerns:

We thank the reviewer for their review, and we have addressed their comments below.

1)      The abstract as written is quite long and detail-oriented, especially with regards to the summary of results. Please shorten it with the aim of describing only the main points of the study and its significance.

We have now made the abstract more concise:

*"Microtopography can be a key driver of heterogeneity in the ground thermal and hydrological regime of permafrost landscapes. In turn, this heterogeneity can influence plant communities, methane fluxes and the initiation of abrupt thaw processes. Here we have implemented a two-tile representation of microtopography in JULES (the Joint UK Land Environment Simulator), where tiles are representative of repeating patterns of elevation difference. Tiles are coupled by lateral flows of water, heat and redistribution of snow, and a surface water store is added to represent ponding. Simulations are performed*

*of two Siberian polygon sites, (Samoylov and Kytalyk) and two Scandinavian palsa sites (Stordalen and Iškoras).*

*The model represents the observed differences between greater snow depth in hollows vs raised areas well. The model also improves soil moisture for hollows vs the non-tiled configuration ('standard JULES') though the raised tile remains drier than observed. The modelled differences in snow depths and soil moistures between tiles result in the lower tile soil temperatures being warmer for palsa sites, as in reality. However, when comparing the soil temperatures for July at 20 cm depth, the difference in temperature between tiles, or 'temperature splitting', is smaller than observed (3.2 vs 5.5°C). Polygons display small (0.2°C) to zero temperature splitting, in agreement with observations. Consequently, methane fluxes are near identical (+0 to 9%) to those for standard JULES for polygons, though can be greater than standard JULES for palsa sites (+10 to 49%).*

*Through a sensitivity analysis we quantify the relative importance of model processes with respect to soil moistures and temperatures, identifying which parameters result in the greatest uncertainty in modelled temperature. Varying the palsa elevation between 0.5 and 3 m has little effect on modelled soil temperatures, showing that using only two tiles can still be a valid representation of sites with a range of palsa elevations. Mire saturation is heavily dependent on landscape-scale drainage. Lateral conductive fluxes, while small, reduce the temperature splitting by ~1°C, and correspond to the order of observed lateral degradation rates in peat plateau regions, indicating possible application in an area-based thaw model."*

2)      I would like to see the authors include a bit more information about some aspects of JULES which currently are not described, as they were not updated for the present study. In particular, how does JULES handle freeze and thaw in a soil column? Do liquid water and ice ever co-exist in the pore space? Is the expansion of liquid water upon freezing accounted for?

We have included the following material in the JULES description in section 2.2:

*"JULES accounts for the latent heat associated with the freezing / thawing of soil moisture using an apparent heat capacity (Essery et al., 2001), while the unfrozen water content of a layer is calculated from the temperature using a relationship derived from minimising the Gibbs free energy. Frozen and unfrozen water therefore can therefore co-exist, and*

*the unfrozen volumetric water content is used to calculate the matric potential of a frozen layer. The decreased density of water on freezing is however not taken into account, nor are any frost heave effects modelled."*

3)        The authors state in line 302 that "Currently, no thermal effects of the pond are included, and the pond cannot freeze..." This seems like a major simplification, particularly as delayed freeze-up beneath ponds is one of the mechanisms which raises mean annual ground temperatures and drives positive feedbacks on permafrost thaw in cold environments. Please discuss this limitation in more detail in your discussion section. Also, please clarify how the presence of a pond that cannot freeze affects simulations in winter. For example, can snow accumulate atop the pond when air temperatures are below freezing?

We now clarify in the implementation description in 2.3.4:

*"Currently, no thermal effects of the pond are included and so the pond is purely a surface water store. This means that the pond cannot freeze and snow accumulation is unaffected, though in future aspects of FLake (Rooney and Jones, 2010) could be used to introduce these processes, in a similar manner to (Langer et al., 2016). The only process other than infiltration affected by there being water in the surface water store is that when the pond is present, bare soil evaporation is switched off, and evaporation from the surface water store is equal to the potential evaporation rate calculated by JULES (Best et al., 2011). "*

*And note in the discussion in 3.3.1 that:*

*"In the previous section, we discussed that the introduction of ponding only has an effect on the level of soil saturation under certain conditions. For instance, if a pond is present, the soil may be fully saturated whether the pond is 1 cm or 1 m deep. Also, as mentioned in section 2.3.4 Ponding and run-on, currently the pond is purely a water store and lacks any thermal properties. This means that while Iškoras sometimes has a simulated pond depth of over 0.5 m, the soil temperatures are no different to if the pond was very shallow. As such we expect the simulated temperatures to be more representative of a part of the landscape with shallow ponding. In future it would be beneficial to include the thermal effects of ponding, as we would expect the additional latent heat of the pond to cause a delay in soil freeze-up (Abolt et al., 2020; Langer et al., 2011), and even the formation of a talik  at polygon sites if larger pond depths are simulated (Yi et al., 2014).*

*This may require revisiting how the surface drainage of the pond is controlled ($f_{pd}$), and possibly additional tiles to ensure that both the wetland and permanently ponded areas of the gridcell are adequately represented. In this study however, Iškoras is the only site for which a persistent pond forms. The other sites form only a temporary pond after snowmelt, so for these sites adding the thermal effects of ponding would not be expected to affect winter freeze-up."*

**Minor Comments:**

Line 116 - Please describe the "top-down" approach of Turetsky and Schneider von Deimling a bit, instead of simply referencing it.

Previously, we had written:

*"When applied on the global scale, a more top-down rates-based approach, similar to that taken by (Turetsky et al., 2020; Schneider von Deimling et al., 2015) may well be necessary due to the complexity of the feedbacks involved."*

This now reads:

*"When applied on the global scale, a more top-down approach to modelling future permafrost carbon fluxes may be necessary due to the complexity of the feedbacks involved. An example of this method of approach is that taken by Turetsky et al. (2020), who collated observed current rates of abrupt thaw, land type areas, carbon inventories and fluxes, and projected these into the future using modelled gradual thaw rates. Similarly, Schneider von Deimling et al., (2015) divide latitudinal bands into different land types with associated carbon pools. Carbon pools and rates for the different land types are set to match observations, and future changes in area are scaled by surface air temperature anomaly from CMIP-5 models under different scenarios."*

Line 240 - What is the "sloped area" of polygonal tundra?

We hopefully have clarified this by writing:

*"for low-centred polygons the soil under the slope between rim and centre"*

Line 416 - This definition of Δx is vague. Please clarify it and describe how you justified your choice of distance.

We have expanded the discussion of the selection of the values for $\Delta x$ as follows:

*"$\Delta x$ is the horizontal distance in the calculation of the thermal and hydological gradient between tiles (Eqs. 5 & 6), and hence a key variable in determining horizontal fluxes of water and heat. $\Delta x$ is chosen to be representative of the length over which the transition of the thermal and hydrological regimes between the two tiles takes place, as it is this that affects the magnitude of the fluxes. $\Delta x$ is therefore typically less than the distance between the centres of each tile. Abolt et al.'s (2020) simulations of low-centred polygons in Prudhoe bay, Alaska, show an approximate distance for this transition only slightly longer than the length of the slope between the edge of the flat centre area and the top of the rim (~1 m). We note that the Prudhoe Bay site has a 30 cm organic layer and a relatively sharp transition between rim and centre. These factors as well as climatic differences mean that $\Delta x$ may be different for our sites. We therefore chose values for $\Delta x$ that are slightly longer than the length of observed transition in topography, noting that these values are similar to those chosen in previous studies (Aas et al., 2016; Nitzbon et al., 2019), but we later test the effect of this assumption in the sensitivity analysis."*

Line 488 - Please elaborate on what qbase is mathematically. Is this water which is extracted from the soil and leaves the model domain completely?

Previously we wrote:

*"JULES uses TOPMODEL to calculate the baseflow (qbase) or 'subsurface runoff' for each soil layer, based on a topographic index and the position of the water table. This acts as the gridcell-scale lateral drainage. Here, we also qbase set to zero for layers which are unsaturated and/or frozen."*

We have now elaborated on this as follows:

*"JULES uses TOPMODEL to calculate the baseflow, the saturated flux laterally exiting the gridcell, based on the position of the water table and the distribution of topographic index within the gridcell as described in Gedney and Cox (2003). The calculation is based on the following assumptions: that the water table follows the gridcell topography, that as the water table rises the baseflow increases due to the increasing transmissivity of the saturated zone, and that in the steady state the downslope flow is balanced by the upslope recharge. In JULES this flux is known as qbase and is calculated for and extracted from a layer that contains the water table or that is beneath the water table. Here, we also set qbase to zero for layers which are unsaturated and/or frozen. The total qbase for*

*the gridcell is passed into the river routing scheme if this is being used, but otherwise passes out of the model domain. Each gridcell does not receive any flux from the gridcells surrounding it, so this can be viewed as the flux balancing the recharge that the groundwater receives in the gridcell area."*

Figure 9 - Please describe the symbology of your box and whisker plots (e.g., meaning of center line, box height, and whisker extent).

We have now included the following in section 3.2.2:

*"In this plot and all following box plots, the box extends from the first to the third quartile with the centre line at the median. The whiskers extend from the box by 1.5 times the interquartile range and fliers indicate years where the mean flux was outside the whiskers."*

Line 653 - I am unsure what "liquid water content as a fraction of saturation" means. Is this averaged among all the layers in a tile? Also, since you use the word "reduce", I think the negative signs in in line 354 are unnecessary and confusing.

In section 3.2.2 we now clarify the shorthand 'fraction of saturation': "liquid water as a fraction of pore space, henceforward referred to as fraction of saturation". We have also clarified in 3.2.3 that we are talking about the "the liquid fractional saturation at 0.19 m", rather than an average of all the layers. This terminology has also been clarified in other places identified by reviewer #2. The negative signs have been removed from line 654.

Line 658 - It is very interesting that turning off heat fluxes between tiles seems to have more impact on tile saturation than turning off water fluxes. Please explain in more detail why this is the case, as it is not intuitive.

The reviewer raises a good point here, but it is hard to point to any one thing as the result of turning off lateral fluxes of heat. In fact the consequence is a small change in several fluxes in several places! We have rewritten the paragraph to give some indication of this:

[revised manuscript text omitted]

Line 887 - Please check whether GMD allows for citation of submitted manuscripts which have not yet been published.

This paper has now been published, so this is no longer a problem.

**Anonymous Referee #2**

In this study, Smith et al. introduce new implementations into the JULES land surface model for the representation of micro-scale heterogeneities in permafrost landscapes. They put a focus on ice-wedge polygons which are common in lowland continuous permafrost and on palsas in the discontinuous/sporadic permafrost zone. The authors

describe in detail the novel implementations which are based on the concept of laterally coupled tiles which was previously introduced for these kind of landscapes by Aas et al. (2019) and Nitzbon et al. (2019). The new model implementations were evaluated using field observations from four sites, and parameter and process sensitivity studies were carried out to assess model uncertainties.

While the scientific concepts the study builds on are not novel, their implementation into JULES, the description of caveats in doing so, as well as the thorough analysis of model sensitivities are valuable contributions for improving the representation of permafrost in Earth system models. This justifies the publication of the article in GMD. However, the article in its present form has several (mostly minor) weaknesses which the authors should address before the article can be accepted for publication. These concern primarily the presentation of the findings and are listed below.

We thank the reviewer for their detailed and helpful review, and we have addressed their comments below.

**General comments**

■ The abstract of the article is quite long and contains a lot of details which are not necessary to be included here. The authors should condense the most important points and keep details for the main text. Similarly, the introduction section is rather exhaustive and should be reduced. For example, the discussion of abrupt thaw processes in [l.99ff] could partly be saved for the outlook section, and the description in [l.121ff] are not necessary in this detail in the introduction.

The abstract has now been shortened (see reviewer #1).

We have swapped the paragraph on abrupt thaw in the introduction for a single sentence in the opening paragraph:

*"Localised thawing can be further exacerbated by a positive feedback due to subsidence from melting ground ice in an abrupt thaw event (Walter Anthony et al., 2018)."*

We have also deleted the following sentences:

*"The two tiles interact via exchange of both soil moisture and heat, by surface run-on and by redistribution of snow. A surface water store is added to the low tile to represent ponding."*

■ I see that the authors aimed for a thorough evaluation of the tiled model configuration by comparing the results to various observational data. The study lacks, however, an evaluation of the model's capability to realistically simulate the thawing of ground, i.e. how the seasonal thaw front propagates and how deep the active layers are. Due to the crucial importance of the thaw depth for a range of other processes in permafrost ecosystems, it would be desirable to also assess the model's capability to simulate thaw depths, and to discuss how these are affected by the tiling scheme. Observational data of active layer depths should be available at least for some of the study sites.

Observed and spatially resolved thaw depths are available for Samoylov and Iškoras. We now show thaw depth on the overview plots (see figures 5 and B6 below), and have added a separate plot (figure 11) along with the following text in section 3.3.1 - The effect of microtopography on soil temperatures:

*"Modelled thaw depths are compared with observations in Figure 11, and were diagnosed by linearly interpolating the position of $0°C$ soil temperature closest to the surface. As with the soil temperatures, tiling causes a clear difference in thaw depths for palsa sites and little difference for polygon sites. It is helpful to compare these plots with the overview plots of soil temperature and unfrozen water in Figure 5 and also Figure B 6 in the appendix. For the mire at Iskoras a clear talik can be seen, with the surface freezing to around 0.5m in winter. For the modelled mire at Stordalen, while the $0°C$ depth only just exceeds 1 m in summer, soil column temperature never drops far below $0°C$ and at 0.5 m depth retains a substantial amount of liquid water year-round. While the tiled thaw depths for Iškoras are much closer to observations than for standard JULES, thawing is somewhat earlier than observed and 22 cm deeper for the high tile. A greater modelled ice content in the palsa, either through excess ice or a greater high tile saturation could perhaps reduce this. For Samoylov, thawing in the model occurs 2-3 weeks earlier than observed. The model also shows a pronounced (~15 cm) decrease in thaw depth between September and October before surface freeze-up, as opposed to the delayed decrease of soil temperatures observed. Again, this corresponds to the earlier snowmelt in spring, and delayed build-up of snow in Autumn compared with observations (Figure 6). This contributes to the maximum thaw depth in September being ~ 15cm smaller than observed."*

[Figure]

*"Figure 11: Climatologies (2002-2016) of modelled vs observed thaw depths. Observed thaw depths for Iškoras were from 2019, Samoylov CALM thaw depths were from 2002 – 2016, grouped according to classes 1 and 3 and averaged using a 3rd degree polyfit across transects."*

▪ I do not quite understand why the snow scheme is evaluated in terms of the „climatology" of snow depths (Figure 7). As the observational data are available for specific years, why are these not compared directly with the respective simulations? The authors write that „the simulation has the correct depth of snow on the rim, but around 20 to 30 cm too much snow on the centre", but this is not visible from Figure 7. Also, there seems to be an issue with the observed „snow depth" in Samoylov during the summer months which is >0cm. Maybe the sensor measured vegetation, but the data should be corrected such that the snow depth is 0 when there is no snow.

We chose to evaluate the snow scheme in terms of the climatology of snow depths to smooth out some of the variability inherent in using reanalysis data to drive our model rather than observed snowfall. Using a climatology makes the average effect on the snow depths easier to see, however for completeness we now include the full timeseries in the appendix:

[Figure]

*"Figure B 5: Timeseries comparing observed and modelled snow depths for Samoylov showing inter-annual variability in snowfall. Snow depths have been bias-corrected using the signal during the snow-free season. Stars denote the median observed value across multiple polygons by Gouttevin et al. (2018) on their campaign in April 2013."*

We have also clarified the points measured by Gouttevin et al:

*"The full timeseries for Samoylov is given in the appendix in Figure B 5, showing the interannual variability between the snowfall reanalysis data used to drive the model and the observed snowfall. Figure B 5 also shows the medians for the rim and centre measured across multiple polygons by Gouttevin et al. (2018) on their campaign in April 2013; for this date the simulation has the correct depth of snow on the rim, but around 20 to 30 cm too much snow on the centre, taking into account the observed spatial variability."*

Observed snow depths for Samoylov have now been corrected for zero error by applying an offset linearly interpolated between the mean 'measured' snow depths in the summer months:

[Figure]

*"**Figure 6:** Climatology (2002-2016) of simulated vs observed snow depths for Samoylov (left) and Iškoras (right), showing snow redistribution due to microtopography differentiating snow depths. Observed snow depths for Samoylov have been bias-corrected using the signal during the snow-free season."*

The authors should invest some effort in improving the quality of the figures (e.g. increase size of axes labels, consistent use of background grids, etc.). In my view, Figure 6 is particularly problematic and – in its current form – fails to provide a good overview of the results which is probably its intention. The entire figure should be thoroughly and carefully revised in order to be insightful. This concerns the placement of the panels (Why not plotting observations and modelling results for each landscape tile next to each other?), the aspect ratio (the five panels in each row have four different aspect ratios which is very irritating), and the design in general (sometimes the plot for the low tile is shifted compared to the high tile, sometimes not...). If the authors like to keep this figure, it would be consistent to provide the respective panels also for the other sites.

We have modified the placement of panels, aspect ratios and axes limits for figure 6 to make the figure more consistent. Panels for the other two sites are now provided in an appendix:

[Figure]

Samoylov

[Figure]

Iškoras

[Figure]

Kytalyk

[Figure]

Stordalen

*"Figure B 6: Climatologies (2002-2016) providing a comparison of observations with standard and tiled (no qbase) JULES for the polygon site Kytalyk and the palsa site Stordalen. The white line above the plots of soil temperature shows simulated snow depth, the cyan line above the plots of simulated soil moisture shows simulated pond depth"*

We have also moved the legend inside figure 10, to allow the panels to take up more space.

[Figure]

**Specific comments**

■ The methods description is the longest part of the paper, which is understandable for a model description paper. However, to my opinion it contains some very JULES-specific information which are possibly not relevant to a broad readership (essentially, section 2.4). I therefore suggest to move these parts entirely to a supplement or an appendix.

We have moved the details of the JULES specific modifications to the supplementary material, and provide only the following brief overview in section 2.4:

*"For this study, we also implemented two modifications to the JULES code that are not of core relevance to modelling microtopography and may not be of relevance to other models that have a different approach to modelling hydrology. Firstly, it was necessary to implement a method of determining the position of the water table within a partially saturated layer, in a way that was consistent with the cell-centred method JULES uses to solve the Richard's equation. This was required in order to determine the local water table for a layer. Secondly, the standard methods JULES uses to avoid supersaturation or undersaturation as a result of the water flux calculation numerics (controlled by the switch*

*l_soil_sat_down) can result in the unintended consequence of water being passed out of the soil column. This is particularly a problem for freezing saturated soils. We implemented a new method which avoids this problem (soil_sat_updown), and which also integrates with the scheme for simulating lateral fluxes of water. These modifications are described in the supplementary material."*

▪ The model setup description is very comprehensive regarding the parameter variations and configurations. However, some information are missing: How is the subsurface stratigraphy set up (ice contents, organic contents, soil texture etc.)? How is the snow represented and how were snow-specific parameters chosen?

We have added the following to the first paragraph of section 2.5.2 Configurations of JULES simulations:

*"JULES snow parameters are the same as those used in the evaluation of UKESM1 by Sellar et al. (2019), with a fresh snow density of 109 kg m$^{-3}$. Soil properties for Iškoras were set up using the same method previously used for the other three sites in Chadburn et al., (2017). For Iškoras the profile was assumed to be entirely peat, which is the case for the first 1.55 m in reality (Kjellman et al., 2018) and is a suitable assumption for our analysis. Excess ice is not currently represented in JULES, so ice contents need not be initialised."*

More information on how JULES calculates the proportion of frozen water in a layer is given in section 2.2, and soil ancillaries are available in the data availability statement.

▪ With few exceptions, I found the figure captions in the results section too long as they not only describe what is displayed but discuss and interpret the data. Such interpretations should be provided in the main text.

We have removed the interpretations from figure captions for figures in the main text and appendices.

▪ At times, I found the language quite technical and loaded with modelling „jargon". Examples: [l.398f] „This does however suggest that l_soil_sat_down = false is in general the more physically realistic scheme." [l.646ff] „In order to directly attribute the effect of

different model processes on soil relative saturation, a series of runs were performed with individual processes switched off in turn with tiled no qbase as the base configuration, the 'subtractive process switching'." Such sentences are very hard to understand in isolation and I suggest to revise the language to more verbal descriptions.

We identified and corrected three main sources of 'jargon', making the manuscript more readable.

The first was the necessarily JULES – specific terminology such as in the discussion of the switch 'l_soil_sat_down'. This is now less of a problem as the section on JULES-specific modifications is now in the supplement, and readers interested in this section would likely be already aware of this terminology. We have however clarified L.396:

*"For the most part, limiting the incoming fluxes to a saturated layer (soilsat-updown) results in a soil moisture profile very similar to that of simply limiting fluxes into the top of a saturated layer (l_soil_sat_down = false), and there are only a very few cases where flux out of the top of the soil is avoided (Figure S 1)."*

The second is in the discussion of qbase, itself a JULES-specific term. We consider it important to stick to discussing the effect of impeding 'qbase' rather than simply replacing 'qbase' everywhere with 'drainage', as 'qbase' is a specific model process in JULES, and this helps distinguish it both from the more general concept of baseflow and from the calculation of lateral flows of water between tiles. However, we have attempted make it clearer to the general reader what is actually going on through the following modifications:

*L579: " -> "Following the discussion in section 2.5.4 on how the landscape-scale drainage calculated by JULES (qbase) should be applied to the tiled model, model results in Figure 7 are shown for both the configuration where this drainage is applied to the mire / rim (qbase on, upper panels), and for the configuration where qbase is switched off for both tiles (no qbase) and the landscape-scale drainage is impeded."*

*L.583: " Impeding drainage from the mire by switching off qbase has a large*

*effect on the palsa sites (right, in orange and yellow), while impeding drainage from rims has a small effect on the polygonal sites"*

*L.587: " -> Conversely, if drainage is not impeded for these sites and qbase is on"*

*L.588: "Impedance of  landscape-scale drainage and limiting qbase in JULES is therefore a necessary condition for modelling palsa mire sites correctly "*

*L.602: "For the rest of this paper,  the tiled configuration with drainage impeded (full tiling - no qbase) is regarded as the best performing model configuration and is the one presented in subsequent figures."*

*L.617: " compared to standard JULES with drainage impeded (qbase off)"*

The third source of jargon is in discussing the model configurations used in the sensitivity study:

*L.656:"In order to directly attribute the effect of different model processes on soil relative saturation, a series of runs were performed  where took the main tiled model configuration (full tiling – no qbase) and switched off individual model processes in turn. These are the 'subtractive process switching' runs referred to in section 2.5.2"*

*L.706: " Figure 12 shows the effect on soil temperatures of the subtractive process switching runs, where we took the main tiling configuration ('full tiling - no qbase', pink) and switched off model processes individually. These are the same runs as shown for soil moisture in Figure 9."*

▪ [Figure 8] This figure is quite loaded and it is hard to distinguish between the individual lines. In particular, the lines for the simulations and observations for Samoylov and Iskoras are hard to distinguish due to the dark colours. In addition, I was confused about

the label at the x-axis. To me, the liquid water content refers to the fraction of liquid water in a given soil volume, which is something else than the fraction of saturation which is the fraction of the pore space filled with water/ice. This should be clarified and the authors should ensure that they are comparing the correct quantities here.

The figure has been updated with brighter colours, and 'Liquid water content (fraction of saturation)' now reads 'Liquid water (fraction of pore space)':

[Figure]

[Figure 9] In order to save space and at the same time facilitate a better comparision amongst the three setups, I suggest to revise the way the data are plotted: Instead of showing four separte boxplots for the three setups, the same information could be shown in one (wide) boxplot where the data for the three setups are plotted directly next to each other, i.e. three boxes for runoff/on, three boxes for qbase, etc. In this way, there might

be even enough space to include the results for the other sites (Figure B3) in the main text. Other than that, it seems that units provided for the yaxis (mm / m^2 / year) are wrong. I suspect this should just be (mm / year).

We thank the reviewer for the great suggestion which we have now implemented. The units have also been corrected:

[Figure]

■ [l.36] Writing „Consequently,…" suggests that methane emissions would be determined mainly by soil temperatures, but an improved representation of saturated parts of the landscape would also affect methane emissions (e.g. from wet polygon centres).

In this case, the word consequently is justified as for our analysis we are effectively only comparing the methane flux per area from the 'wetland fraction', so fluxes are not directly affected by the water table depth (as discussed in the introduction and at the end of section 3.4). As discussed in section 3.4 the direct effect of water table dynamics on methane production would likely add to the temperature feedback, but would require further consideration as a major change to how methane fluxes are calculated in JULES.

■ [l.63] A reference should be provided.

This sentence now reads:

*"While rates of methane emissions from anaerobic decomposition are generally much lower than those of aerobic decomposition and the production of $CO_2$, the high Global Warming Potential (GWP) of methane means the former could be of comparable importance to the permafrost carbon feedback in the short term (Walter Anthony et al., 2018; Turetsky et al., 2020; Schneider von Deimling et al., 2015)."*

[Figure 1] A scale for reference should be included in both pictures. If possible, hte picture for the palsas should be replaced by an aerial image in order to motivate the assumption of „repeated patterns" (e.g., https://www.norgeskart.no/ provides high-res imagery).

We have now included an aerial image for Iškoras, and an additional aerial image for Samoylov, both containing a scale. As we do not have a measurements for the displayed polygons and palsas in the close up images, we have included average measurements from the sites in the figure caption:

[Figure]

*"Ice-wedge polygons at Samoylov (left, J. Boike) and palsas and mire at Iškoras (top right ©Kartverket, norgeskart.no, lower right, N.D. Smith) showing repetitive microtopography and some surface ponding. The observed low centred polygons at Samoylov were on average 9.1 m in diameter and had rims 0.38 m above the centre. The observed palsa at Iškoras was 0.68 m above the mire. Some polygon rims can reach 1 m in height while Palsas can reach larger elevations of around to 3 m."*

[l.111ff] Here, the works of other permafrost modelling groups, e.g. conducted within the scope of the NGEE Arctic project, could be worth mentioning as well.

We have now included references for some other modelling groups who have tested their model against spatially resolved observations at NGEE sites:

*"Modelling efforts have been hampered by a lack of long-term high-resolution observations, nevertheless, at a handful of field sites the local variability has been well quantified. Data from these sites has been used to test both high resolution models of individual features (Jan et al., 2020; Kumar et al., 2016; Grant et al., 2017), and also low resolution 'tiling' approaches to modelling micro- and meso-topography (Langer et al., 2016; Aas et al., 2016, 2019; Nitzbon et al., 2019, 2020a; Cai et al., 2020; Martin et al., 2021). The reduced computational burden of the latter approach has been pursued as a possible route to representing sub-grid processes in ESMs."*

[l.407] I think „ice segregation" would be the correct process as frost heave is a more general phenomenon.

Good point, this has been changed.

[l.415] The formula for seems to be for the polygon centre area in Figure 5 (A_2 instead of A_1. But considering the explanations in ll.421ff, the approach was to determine A_1 and A_2 independently, and to obtain $\Delta l$ based on these.

Well spotted, this should read $A_2$ and has been changed. We have attempted to clarify l.422 as follows:

*"Averages were taken over the major and minor axes of each polygon to calculate an average area for the centre. An average of measurements of rim width were then used to calculate the area of an annulus about the centre representing the rim."*

[l.416ff] It is not clear how the parameter $\Delta x$ was determined. If is is calculated based on the areas and/or perimeters, a formula should be provided.

We have expanded the discussion of how $\Delta x$ was determined (see comment by reviewer #1).

[l.486] „Work to improve ..." I find such statements irrelevant for the present work.

This sentence has been removed.

[l.487ff] I appreciate the discussion on how to model baseflow as it addresses general issues arising when representing micro-scale processes within large-scale LSMs. However, there is no real-world motivation provided for the „twin qbase" scenario, and I think it was not discussed explicitly in the Results section. The authors could consider discussing these scenarios in the Discussion section, especially in the context of changing hydrological connectivity, e.g. through ground subsidence.

We have now included the following at the end of the discussion of impeding qbase in section 3.2.1:

*"So far we have discussed whether qbase should be applied to one of the tiles or to neither of the tiles. Applying qbase to both tiles (twin qbase) was not considered originally because of the assumption that the polygon rim surrounds the polygon centre, and likewise the mire around the palsa. However, there may be situations where some qbase from both tiles is possible, such as if subsidence causes a breach in the polygon rim, or in situations where there is not such a well-defined and regular combination of palsa and mire. By comparing the configuration where qbase is applied to only to the mire / rim (full tiling - qbase on) with the twin qbase configuration in Figure 9 (discussed in more detail later) we can see what effect also calculating qbase from the polygon centre / palsa has. For the latter, we find that there is almost no effect on the palsa saturation. This agrees with what we see in Figure 7, where the lateral fluxes from the palsa cause a saturation profile similar to that of the mire with qbase. For polygons there is a reduction in liquid water as a fraction of pore space for July at 0.19 m of 0.21 for Samoylov, and 0.05 for Kytalyk. This is a bigger effect than switching off snow redistribution for Samoylov, although not for Kytalyk, and for both sites the low tile is still significantly wetter than standard JULES. In sites with high hydrologic connectivity, we therefore still expect to see effects of microtopography on soil saturation where thaw depths are shallow."*

[l.507ff] The lateral landscape-scale drainage would most likely also be influenced by the meso-scale topography [e.g. Nitzbon et al., 2021].

This line now reads:

*"In reality, the lateral landscape-scale drainage will depend on a combination of the meso-scale topography (Nitzbon et al., 2021), the connectivity of the polygon troughs or mire network (Liljedahl et al., 2016; Connon et al., 2014), and the presence of external reservoirs and inter-gridcell flows."*

[l.591] The daily external water fluxes stated in the model of Martin et al. (2019) are only applied when the ground surface is unfrozen. Hence, the calculated annual fluxes are only „potential" fluxes. The actual fluxes „per thawing season" would be considerably smaller. This should be clarified as well as revised how these number compare to the annual qbase fluxes in JULES which they are compared to.

Thank you for notifying us of this, the line now reads:

*"This agrees with the simulations of Martin et al. (2019), who when varying the external water flux in their peat plateau model found a "drainage effect" when transitioning from an external influx of +1.5 mm/day during the thawing season (360 mm/year) to an outflux of -2 mm/day during the thawing season (-380 mm/year). Here, to get an approximate value of total annual flux from the daily flux during the thawing season, we have multiplied by the days unfrozen recorded by the loggers Su-L14 and Su-L4 for the wet mire and dry palsa respectively."*

This doesn't change the rest of the analysis, however we clarified l.600 as follows:

*"In fact, very little drainage is required for the mires to become unsaturated ($fd_2 \sim 0.2$), which corresponds to a flux closer to the middle of the transition found by Martin et al."*

▪ [l.747] The authors write: „[…] explicitly modelling microtopography can increase modelled methane fluxes […] in some cases." It should be noted that these cases are exactly those sites where recent permafrost degradation is strongest, i.e. regions which are transitioning from permafrost- to non-permafrost conditions.

This is a good point, though we thought it might be better discussed in the summary and outlook section. We have changed l7.4.7 to be more specific:

"These results show that explicitly modelling microtopography can increase modelled methane fluxes vs the standard JULES in cases where microtopography driven processes enable sufficiently different soil thermal and hydrological conditions to coexist."

We have also inserted the following paragraph after the first paragraph of section 4 – summary and outlook:

"Palsa mires are by nature in marginal locations where the transition between permafrost and non-permafrost can be seen. That explicitly modelling microtopography has a greater effect on the modelled fluxes at these sites is unsurprising due to the distinct nature of

these conditions which cannot coexist in standard JULES. Future studies should investigate the potential difference to the modelled area of permafrost and methane production when a tiled approach to microtopography is applied at the pan-arctic scale. The marginal nature of palsa mires also makes them sensitive to climate fluctuations (Seppala, 2006). Palsa degradation has been observed at these two sites and more broadly across Europe and western Siberia (Kirpotin et al., 2011; Borge et al., 2016), with one study projecting that over half the area currently suitable for palsas in Fennoscandia is very likely become unsuitable by the 2030s (Fronzek et al., 2010). The changing climate in these areas could be increasing the difference in conditions between permafrost and mire while palsas are still present but are being pushed out of their climate envelope of stability. This motivates modelling microtopography for the purpose of understanding the future effects on methane fluxes of new areas experiencing permafrost degradation."

▪ [Figure A1] To my opinion, these plots are not necessary, as long as the depths and times for which model and observations are compared are stated clearly in the main text or the main figures.

We have removed the figures showing observation depths and date ranges. The following sentence has been added to the figure captions comparing results of simulated soil temperature at 0.19 m depth with observations:

*"Depth of observations: Samoylov - rim 0.21 m, centre 0.2 m; Kytalyk – rim 0.15 m, centre 0.25 m; Iškoras – palsa and mire 0.2 m; Stordalen – mire 0.25 m."*

Similarly, the following sentence was added to figure captions comparing simulated soil moistures at 0.19m with observations:

*"Depth of observations: Samoylov - rim 0.22 m, centre 0.23 m; Iškoras– palsa and mire 0.2m."*

*To avoid repetition, the following sentence and table was added at the start of section 3: "Date ranges of available soil temperature and moisture measurements are given in Table 4."*

| | Samoylov | Kytalyk | Iškoras | Stordalen |
|---|---|---|---|---|
| Soil moisture & temperature | January 2002 - December 2019 | January 2012 - December 2015 | January 2017 - September 2019 | Palsa: June 2006 - January 2015 Mire: September 2005 - September 2012 |
| Thaw depth | 2002 - 2018 | -- | 2019 | -- |
| Snow depth | 2012 - 2019 | -- | 2018 - 2020 | -- |

■ [Figure B1/B2] I found it confusing that for the soil moisture splitting, the subtractive/additive process switching results are shown in the main text/appendix (Figure 10/B1) while it is the other way around for the soil temperatures (Figure 12/B2). Even if the results shown in B1/B2 are not discussed in detail, it would be more instructive and allow for better comparison to provide these as part of Figures 10/12 in the main text, e.g. as an additional row for each site. Instead, the tables with the specific numbers could be moved to the appendix as the most important ones are mentioned in the text.

We now show the subtractive version of the soil temperatures in the main text so this is consistent with the soil moisture plot, and also now primarily discuss this version in section 3.2.2. However, we decided that having the additive versions of soil moisture and temperature together in the appendix actually works well, as these plots can now be used to compare how changes to tile saturation affect changes to soil temperatures. It also avoids the reader being overwhelmed with lines for what is now only a small discussion.

■ [Figure B3 and B4] As mentioned above, the figures could be condensed to one panel/plot per site if the respective fluxes for the five setups are plotted next to each other. Otherwise, the axes labels and captions are much too small to be readable without zooming in. Please revise and increase the label sizes.

This has now been implemented and the figures are hopefully more legible.

[Figure]

*Figure B 3:  Average yearly simulated moisture fluxes for the period 1950 to 2016 for all sites, with qbase on. Fluxes for the qbase off configuration, as in Figure 8, are shown in grey for comparison.*

[Figure]

*Figure B 4:  Average yearly simulated energy fluxes (1950 to 2016) for all sites, with qbase on. Fluxes for the qbase off configuration are shown in grey for comparison.*

**Technical corrections**

[l.22] should be „... the model's sensitivity to its parameters."

This line is no longer present in the condensed abstract.

[l.136] „the gridbox average values": Please specify which average is meant here.

We have now clarified that:

"In standard JULES (left), the gridcell consists of a single soil column representing the average values of soil temperature and carbon. These values are used to calculate a

methane flux which is then multiplied by a wetland fraction (diagnosed by TOPMODEL using the modelled water table depth and the gridcell topographic index)."

Similarly in l.66:

"JULES calculates methane fluxes using the values of the soil temperature and carbon pools for each layer (Clark et al., 2011; Gedney et al., 2004). Since JULES is usually run with a single soil column for each gridcell, these values represent the average values across the gridcell."

[l.386] delete „the"

Done.

[l.489f] „we also qbase set to zero for …" Please correct.

Done.

[l.510] correct typesetting for fd1 and fd2 to be consistent with other occurrences.

Done.

[l.663] Please write „standard" instead of „std".

Done.

[l.813] „choose" instead of „chose"

Done (assuming this was for l.831).

[l.865] I think this should be „through" instead of „though".
Done.

**Gautam Bisht**

In this study (Noah et al. submitted; hereafter N2021), the standard JULES model is extended by implementing a two-tile representation of microtopography (JULES vn5.4_microtopography) that accounts for lateral flow of water, heat, and snow redistribution. The new model was validated at permafrost landscapes that included two polygonal and two palsas study sites. The model was able to accurately simulate the difference in snow depth between hollows and rims. Methane fluxes were estimated for the standard and vn5.4_microtopography versions of JULES using observed soil carbon

profiles. While the difference in simulated methane fluxes for the two model versions of JULES was small for polygonal study sites, the difference was large for palsa sites. Additionally, parametric sensitivity analysis showed that the elevation difference parameter for the palsa sites had an insignificant impact on the simulations. However, the exclusion of lateral flow of water and energy modified simulation of soil saturation and soil temperature.

Previously, Bisht et al. (2018) implemented snow redistribution and lateral transport of subsurface hydrologic and thermal processes in the E3SM Land Model (ELM)-3D v1.0. The model simulations were performed for a transect across a polygonal study site in Alaska that is characterized by low-centered polygons. The inclusion of snow redistribution led to a significant reduction in the bias of the difference in snow depth between the polygon center and rim, in a manner similar to that found in N2021. The model was also able to accurately capture warmer winter soil temperature for the center than the rim because of higher thermal insulation from a larger snowpack in the polygon center, again similar to the results in N2021. Finally, the spatial variability of soil moisture and temperature were overestimated in the ELM-3Dv1.0 simulation that excluded lateral transport of water and energy.

Given the very strong relevance of Bisht et al. (2018) to the N2021 study and analogous conclusions for aspects of the results, it would be beneficial if the authors discussed the differences and similarities of their results with those found in Bisht et al. (2018).

*centred polygon site in Barrow, Alaska. They found that the active layer depth (ALD) was ~ 10 cm shallower under rims and ~ 5 cm greater under centres with snow redistribution on vs the standard simulation. When lateral flows were also turned on (physics = 2-D), ALDs were ~ 7 cm deeper under rims and ~ 2.5 cm shallower under centres than the standard simulation. Atchley et al. (2015) used a 1D version of the Arctic Terrestrial Simulator (ATS) also at Barrow that found ~ 3 cm deeper thaw depths in centres and ~ 0.3 cm deeper thaw depths in rims with snow redistribution turned on. In our simulation, we found ALDs for the tiled simulation were on average 1.1 and 6.1 cm deeper for the rim and 0.1 cm shallower and 4.3 cm deeper for the centre for Samoylov and Kytalyk respectively. While our simulations are not at Barrow, we note that our differences in the thaw depths for Samoylov are particularly small compared to the results of Bisht et al. (2018) and Atchley et al. (2015). We also see that together these authors similarly find that thaw depths in polygon centres can become shallower or deeper when microtopographic processes are switched on. In a similar manner to the smoothing effect of subsurface processes found by Bisht et al., in the sensitivity study in the next section we find that while snow redistribution causes colder high tiles and warmer centres, lateral flows of heat mean that much of this difference is cancelled out in summer. We also find that our choice of $\Delta x$ is in a local minimum for Kytalyk, such that a small increase or decrease can lead to one tile or the other being warmer and having a greater ALD in summer."*